

# Inland migration of near-surface crevasses in the Amundsen Sea Sector, West Antarctica

Andrew O. Hoffman[1], Knut Christianson[2], Ching-Yao Lai[3,4], Ian Joughin[5], Nicholas Holschuh[6], Elizabeth Case[1], Jonathan Kingslake[1], and the GHOST science team[*]

[1]Lamont-Doherty Earth Observatory, Columbia University, Palisades, NY, USA
[2]Department of Earth and Space Sciences, University of Washington, Seattle, WA, USA
[3]Department of Geosciences, Princeton University, Princeton, NJ 08544, USA
[4]Department of Geophysics, Stanford University, Stanford, CA 94305, USA
[5]Polar Science Center, Applied Physics Laboratory, Seattle, WA 98105, USA
[6]Department of Geology, Amherst College, Amherst, MA 01002, USA
[*]GHOST is a part of the International Thwaites Glacier Collaboration. For more information regarding the team, please visit the link which appears at the end of the paper.

**Correspondence:** Andrew Hoffman (aoh2111@columbia.edu)

**Abstract.** Since distributed satellite observations of elevation change and velocity became available in the 1990s, Thwaites, Pine Island, Haynes, Pope, and Kohler Glaciers, located in Antarctica's Amundsen Sea Embayment, have thinned and accelerated in response to ocean-induced melting and grounding-line retreat. We develop a crevasse image segmentation algorithm to identify and map surface crevasses on the grounded portions of Thwaites, Pine Island, Haynes, Pope, and Kohler Glaciers between 2015 and 2022 using Sentinel-1A satellite synthetic aperture radar (SAR) imagery. We also develop a geometric model for firn tensile strength dependent on porosity and the tensile strength of ice. On Pine Island and Thwaites Glaciers, which have both accelerated since 2015, crevassing has expanded tens of kilometers upstream of the 2015 extent. From the crevasse time series, we find that crevassing is strongly linked to principal surface stresses and consistent with von Mises fracture theory predictions. Our geometric model, analysis of SAR, and optical imagery, together with ice-penetrating radar data, suggest that these crevasses are near-surface features restricted to the firn. The porosity dependence of the near-surface tensile strength of the ice sheet may explain discrepancies between the tensile strength inferred from remotely-sensed surface crevasse observations and tensile strength measured in laboratory experiments, which often focus on ice (rather than firn) fracture. The near-surface nature of these features suggests that the expansion of crevasses inland has a limited direct impact on glacier mechanics.

## 1 Introduction

Crevasses, common to almost all glaciers, are the result of macroscopic material failure due to the progressive accumulation of micro-cracks and voids in glacier ice (Schulson et al., 1989; Vaughan, 1993; Veen, 1999; Colgan et al., 2016). These cracks and inclusions can affect glacier mechanics by promoting iceberg calving (Berg and Bassis, 2022), changing the bulk density and viscosity of glaciers (Meier et al., 1994), and efficiently routing water to the englacial and subglaical hydrology systems





(Chudley et al., 2021). Despite the connection to englacial stress and water storage and the implications for ice-sheet mass balance, the physical properties of ice that control the mechanics of fracture remain poorly constrained across the Greenland and Antarctic ice sheets (e.g. Bassis and Walker, 2012; Ultee and Bassis, 2016; Alley et al., 2023). As a result, ice-sheet fracture has been suggested as one of the larger sources of uncertainty in projections of future ice-sheet mass loss in the next century (Alley et al., 2023).

Crevasses typically form near the floating margins of ice sheets where horizontal strain rates are high (Lai et al., 2020). Here, simple empirical relationships between iceberg calving rate and horizontal ice divergence suggest a strong relationship between tensile strain rates and full-thickness crevasse penetration (Alley et al., 2008). These observations agree with laboratory experiments that measure the strain response of ice under different applied tensile stresses to determine the critical tensile strength, the critical tensile stress beyond which ice fractures.

In laboratory experiments, the tensile strength of ice depends strongly on ice temperature. For temperatures common in Earth's ice sheets, ice fails when effective stresses exceed a threshold between 1 and 3 MPa (Haynes, 1979). Similar experiments conducted on loose snow to understand the failure behavior of avalanche slabs have found that snow crystal lattices fail under significantly less stress (as low as $\sim 10$ kPa,  McClung, 1978). Tensile strengths inferred from satellite observations of strain rates where crevasses have been manually identified lie between these experimental results on ice and snow, typically

between 100 kPa - 200 kPa (Vaughan, 1993). Together, these satellite and laboratory observations suggest a complicated and multivariate relationship between tensile strength and independent variables, such as density, temperature, and grain size.

Understanding the material strength of ice and its dependencies is important to accurately simulate fracture behavior that may change with the englacial stresses of the ice sheet as terminus conditions evolve under future climate forcing. When surface meltwater fills surface crevasses, the additional hydrostatic pressure of the water can cause the crevasses to penetrate

deeper through a process known as hydrofracture (Nye and Perutz, 1957; Banwell et al., 2013). Over the last three decades, surface melting has intensified in both Antarctica and Greenland, with longer melt seasons extending further poleward (Cook and Vaughan, 2010). Some models predict that warming will increase meltwater-driven hydrofracture in the ASE leading to ice-shelf breakup (via the hydrofracture mechanism) and the formation of ice cliffs that collapse when the newly unbuttressed ice cliff exceeds a critical height. Together, these two processes promote rapid grounding line retreat, termed the marine ice-cliff

instability (MICI) hypothesis (DeConto and Pollard, 2016). The MICI hypothesis predicts that, if progressive failure exposes increasingly taller terminus cliffs, that could initiate runaway collapse. Because the ice sheet in the ASE sits on a glacier bed that deepens into the interior, terminal retreat by fracture could promote taller cliffs and runaway retreat (DeConto and Pollard, 2016). Follow-on work has shown that the elastic response of the ice cliff (Clerc et al., 2019) and the effects of melange buttressing (Schlemm et al., 2022) impact the rate of retreat, likely preventing unstable MICI retreat altogether (Needell and

Holschuh, 2023). Reduced model experiments (Clerc et al., 2019) and 3D discrete-particle modeling have also shown that the mechanics of the MICI depend critically on the tensile strength of ice (Crawford et al., 2021), estimates of which, as noted above, disagree significantly between satellite observations and laboratory experiments. To date, observed changes in surface crevasse appearance remain disconnected from an assessment of ASE glacier vulnerability and predictions of outlet glacier response.





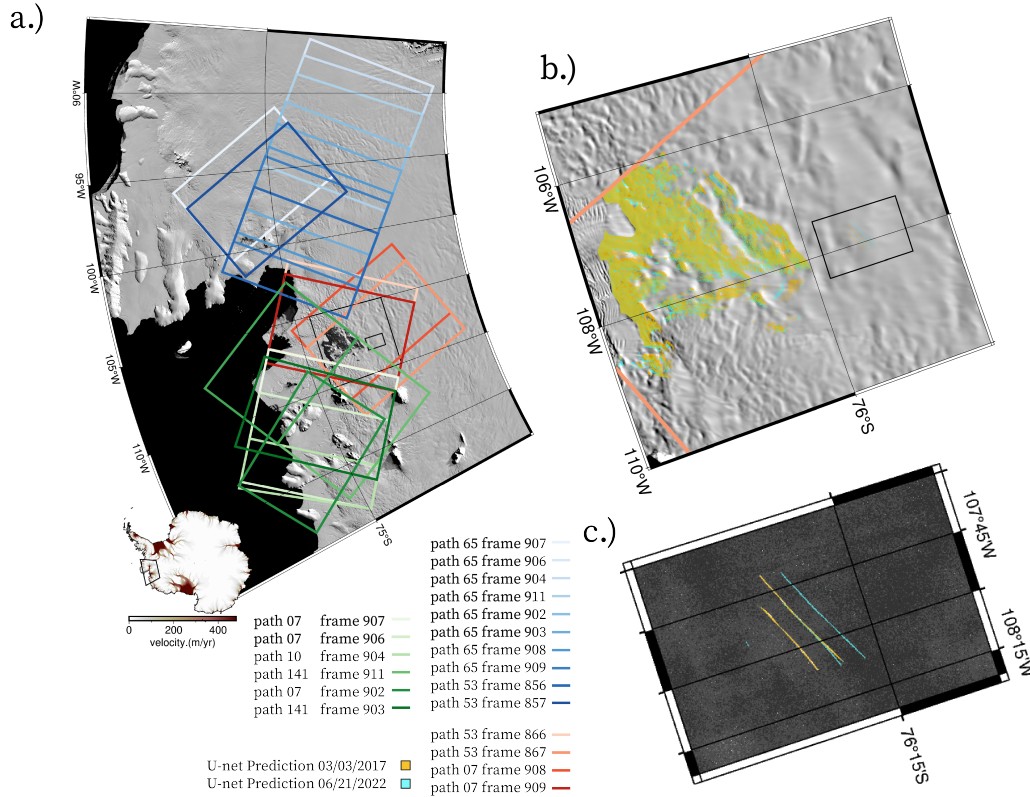

**Figure 1.** (a) Overview of Sentinel-1A scenes of Amundsen Sea glaciers used to create inland crevasse area time series for the grounded regions of Pine Island (lavender), Thwaites (blue, orange), and glaciers feeding the Dotson and Crosson ice shelves (green and red). Black boxes mark areas of paneled images in (b) and (c). (b) An example of the map of crevasses predicted by the U-net from March 2017 (cyan) and June 2022 (orange). Note that the density of crevasses appears to be concentrated on the stoss side of ridges that are captured in the background MODIS image (Haran et al., 2018). The neural network also skillfully detects (c) isolated crevasse features in low-strain environments.

In this paper, we seek to use the presence and extent of crevassing in the ASE, understood through the lens of fracture mechanics, to estimate the material strength of ice in regions where we observe fracture. In the process, we reexamine how to interpret these features, and seek to determine whether the extent and depth of these features make them a useful proxy for the integrity of damaged ice, as has been interpreted in previous studies (i.e. Surawy-Stepney et al., 2023a).

To map crevasse locations and determine rates of crevasse area change, we extend deep-learning neural network frameworks
(i.e., Ronneberger et al., 2015; Akeret et al., 2017; Lai et al., 2020; Surawy-Stepney et al., 2023a, b) to satellite synthetic aperture radar (SAR) imagery collected by the Sentinel-1 constellation from 2015–2022. Using Worldview optical imagery and ultra-wideband ultra-high frequency radar data we verify the features we identify with our neural network and construct a time series of crevasse location and area change from 2015-2022 across the ASE. We use this time series of crevasse locations



with surface velocity data to test a phenomenological relationship between surface stress and crevasse formation: that crevasses
are present when the surface stresses exceed the material tensile strength (Nye, 1959; Vaughan, 1993; Grinsted et al., 2023, and
citations therein). We then develop a simple geometric model for the tensile strength of polar firn to aid the interpretation of
our observations. Together, our observations and modeling suggest that the presence of surface crevasses in the Amundsen Sea
is restricted to the near-surface porous firn layer, which has a tensile strength that is substantially lower than the tensile strength
of non-porous ice measured in laboratory experiments. We finish by exploring the implications of near-surface crevassing on
marine ice-sheet vulnerability and estimates of mass balance in the ASE sector.

## 2  Data and Methods: From satellite SAR images to maps of crevasse area change

Our investigation comprises four steps. First, we preprocess SAR images and use manually identified crevasse datasets to train
the neural network. We then apply the network to a set of Sentinal-1A images that includes all crevassed areas in the ASE. We
then use surface crevasses visible in contemporaneous optical imagery and ice-penetrating radar data to independently verify
the segmentation algorithm and constrain the depth of the crevasse features. Finally, we use crevasse locations with satellite
observations of surface strain rates to develop a relationship between the effective stress of the ice-sheet surface and the opening
of surface crevasses.

### 2.1  Sentinel SAR Imagery Processing and Analysis

#### 2.1.1  Imagery Selection and Processing

Crevasses can be difficult to detect in optical imagery due to snow bridges. Near-surface crevasses covered by snow bridges
can, however, be mapped remotely using active microwave synthetic aperture radar (Marsh et al., 2021). Compared to optical
imagery, SAR imagery is advantageous because it can provide regular observations through clouds, image features during the
polar night, and penetrate through snow and firn ($> 10$ m for C-band SAR in polar firn; Rignot et al., 2001). In this study, we
use synthetic aperture radar data collected as part of the European Space Agency's Sentinel-1 missions.

Sentinel SAR 1A and 1B Level-1 Ground Range Detected, interferometric, wide-swath, high-resolution, amplitude images
($\sim 5$ m x 20 m native resolution, HH polarization) were selected from 2015 to 2022 using the Alaska Satellite Facility Vertex
search tool (ASF DAAC, 2014-2022). Images were chosen geographically to capture areas that contained the approximate
inland crevasse extent between 2015 and 2022. We processed imagery for the entire Sentinel subset, but our time-series analysis
focuses on 2016 to 2022 due to changes in scene geometry during the satellite commissioning stage and a melt event in 2015
that substantially affected radiometric backscatter amplitude (Ghiz et al., 2021). For geometric consistency, solely Sentinel
1A frames were used (see Figure 1). Sentinel SAR data are available with a repeat time of 12 days; shifts in the acquisition
plan can, however, result in less frequent acquisitions for an individual frame. Images with HH polarization were selected
because they appear to have the advantage of higher backscatter amplitude than VV polarization, which reduces random noise



in preprocessed images (Marsh et al., 2021). Images in both ascending and descending orbits were used to assess possible

differences in detectability based on satellite image look angle and crevasse orientation.

Image processing was conducted using the European Space Agency (ESA) Sentinel Application Platform (SNAP) Toolbox (SNAP, 2022). First, orbit state vectors in the gridded product metadata were updated using precise orbit files supplied by the ESA Global Navigation Satellite System Hub. Several radiometric processing steps were then applied, including removal of antenna thermal noise, border noise removal which masks null pixels introduced in the creation of the ground range detected

products, speckle filtering, and radiometric calibration to backscatter coefficient using Look Up Tables that apply a range-dependent gain including an absolute calibration constant. We applied an ellipsoid correction geolocation-grid operator to convert each image to an orthorectified polar stereographic projection (EPSG:3031) using Sentinel ground-control tie points. Orthorectified images were then resampled to 25 m resolution using a nearest-neighbor interpolation to preserve sharp features. Following radiometric calibration and orthorectification, backscatter coefficients (0 to 1) were converted to unsigned 8-bit

integers (0 to 255) and exported as GeoTIFF files (Supplementary Fig. S1).

### 2.1.2    Neural Network architecture

Pre-processed Sentinel images were then used to train a U-net image segmentation algorithm (Ronneberger et al., 2015; Akeret et al., 2017; Lai et al., 2020, Supplementary Fig. S2). The U-net architecture extends the conventional Convolutional Neural Network (CNN) framework by adding expansive operations (upsampling layers) to the contracting operations (downsampling

layers) to identify and localize features in an image. The downsampling path followed by the upsampling path resembles a U-shape and gives the network its name.

### 2.1.3    Neural Network Training

Training data consists of pairs of images and binary classifications for crevassed and crevasse-free regions. Training data were generated using image masking software written for this purpose as part of this study. The masks were generated by first using

a backscatter reflection coefficient threshold to define an initial binary classification – brighter reflections are often associated with crevasses. This binary classification was then refined by manually selecting crevasses that were too dim to be identified by the threshold and deselecting false-positive features. The image training dataset includes images from only grounded ice areas of ASE glaciers because we found basal crevasses can complicate feature segmentation over ice shelves. These training scenes span the entire time series to capture seasonal and interannual changes in the surface reflectivity potentially induced by

melting and snow metamorphism. The training dataset was augmented by rotating and flipping each image and ground truth mask to produce over one thousand training images from binary classifications of 144 images.

We train the network using these refined binary classification images of crevasses. We implement the U-net segmentation algorithm using the Python package Tensorflow (Abadi et al., 2015) with training data from Thwaites, Pine Island, Haynes, Pope, and Kohler Glaciers. The size of raw Sentinel 1A images ($\sim 17000 \times 25000$ pixels; $> 500$ MB) is too large to segment

at the native image resolution. We, therefore, divide each image into $500 \times 500$ pixel patches and run the U-net algorithm on each patch individually using a sliding window with an offset equal to the output layer image (460 pixels in both dimensions).





We tested a variety of optimizers and found the Adams optimizer maximized the area under the curve (AUC) of the receiver operating characteristic (ROC) that we used as the evaluation metric for model selection. The AUC metric measures the ability of a binary classifier to distinguish between classes and is used as a summary statistic of the ROC curve. The output of the U-net model is a $460 \times 460$ pixel image of the likelihood (ranging from 0 to 1) that a pixel contains part of a fracture. We then used the $F_1$ score, which assesses the predictive skill of a model by elaborating on its class-wise performance to choose a threshold for binary classification (model prediction probability greater than 0.8 maximized the $F_1$ score).

### 2.1.4 Neural Network Application

We apply the trained network on more than three hundred Sentinel-1A SAR images from 20 scenes (Fig. 1a) that were selected for their comprehensive spatial and temporal coverage of lower Thwaites Glacier, Pine Island Glacier, and Haynes, Smith, Pope, and Kohler Glaciers that feed the Dotson and Crosson Ice Shelves. Each scene is divided into $500 \times 500$ pixel patches using the same division scheme implemented for the training data. These images were then combined by merging the images, using the maximum value of the pixels within the shared windows. Because translational equivariance is not preserved in CNNs, clipping schemes are necessary to accurately stitch patches of the segmented image back together. We evaluate a variety of padding schemes and find that padding each segmented layer by 20 pixels reduces the error most in the final image relative to hand-drawn masks.

In agreement with the findings of Marsh et al. (2021), crevasses manifest differently in Sentinel imagery as a function of satellite look angle, the slope of the surface, and crevasse orientation relative to the satellite track (Fig. 3). In descending scenes over Thwaites and Pine Island Glaciers, the left column in Figure 3, crevasses that fracture along the grid north-south axis appear highly reflective; while in the ascending scenes, (the middle column of Figure 3), crevasses that are oriented in the grid east-west direction return the most energy. Segmented crevasse features from single scenes are still meaningful when compared to scenes from the same path and frame, but analysis of a single frame may systematically miss features that are not visible because of the crevasse orientation relative to the sensor look angle. This complicates efforts to compare the segmented crevasses with surface strain rates. To mitigate bias introduced by the effects of satellite look-angle, we combine ascending and descending scenes shot within 30 days of one another and create merged predictions that capture multiple look angles. Image merging was done using the maximum of the union of predictions from individual paths and frames.

### 2.1.5 Validating the network and image filtering

The network was applied to the complete time series and used to calculate the area of the crevassed ice-sheet surface using a 3-month rolling median filter and a threshold of 10 km$^2$ to define outliers to be excluded from the time series. These outliers typically overestimated crevasse-area, likely due to surface melting or surface temperature anomalies that temporarily increase the surface backscatter (Joughin et al., 2016). Because C-band SAR imagery penetrates up to several dozen meters into the subsurface (Rignot et al., 2001), it is difficult to determine whether the crevasse void starts at or below the surface from SAR imagery alone.





To better determine the connection of the features we identify in SAR imagery to the surface appearance and subsurface
geometry of crevasses, we use complementary high-resolution optical imagery (WorldView) and ice-penetrating radar data
collected as part of Operation IceBridge (OIB) and the International Thwaites Glacier Collaboration (ITGC) aerogeophysical
surveys, (Fig. 4). WorldView high-resolution optical imagery does not penetrate into the subsurface and thus presents an
unambiguous image of the crevasse surface geometry. The geometry of crevasses in the subsurface was interpreted using
ice-penetrating radar data (i.e. Nath and Vaughan, 2003; Williams et al., 2014). Interpretations of crevasse geometry in ice-
penetrating radar data are complicated by the dependence of the return on the orientation of the radar profile relative to the
geometry of the crevasse and the generally complex shape of the crevasse itself. The ultra-high-frequency ultra-wideband (600-
900 MHz) radar system flown as part of OIB and ITGC surveys is able to image englacial structures through a kilometer of ice
with a vertical (Rayleigh) resolution that captures changes in conductivity due to density contrasts in the near-surface on the
scale of less than a meter. The four-element antennas transmit a $400$ W pulse into the surface that is then recorded using a 12-bit
analog-to-digital converter (Karidi, 2018). The trace spacing for this instrument is large (pulse repetition frequency of 50kHz
traveling on a platform moving 80 meters per second) compared to the trace spacing of ground-based impulse radar surveys
that have historically been used to map subsurface crevasses, but the instrument is still capable of mapping the subsurface at $5$
m horizontal resolution.

### 2.2   Stress analysis from velocity data

Crevasse occurrence can be related to surface velocity and strain rates using the constitutive relation for ice and assuming that
crevasses form if surface stresses exceed a failure criterion. The most common formulation for fracture nucleation and tensile
failure is the von Mises (octahedral) failure criterion. This criterion assumes that the material can only support an octahedral
shear stress that is less than the tensile strength of ice ($\sigma_t$) and can be formulated as (Vaughan, 1993):

$$\sigma_e < \sigma_t \tag{1}$$

$$\sigma_e^2 = \sigma_{1surf}^2 + \sigma_{2surf}^2 - \sigma_{1surf}\sigma_{2surf} \tag{2}$$

where $\sigma_e$ can be interpreted as the effective stress and $\sigma_{1surf}$ and $\sigma_{2surf}$ are the principal surface stresses parallel to the ice
surface. Because the top boundary of an ice sheet or glacier is a free surface, normal stress and traction are zero at this boundary,
and thus one of the three principal stresses must be normal to the surface and have a magnitude equal to zero (Vaughan, 1993;
Veen, 1999). The other two principal stresses are parallel to the surface. We combine our new crevasse location time series
with satellite SAR observations of ice velocity from the same satellite constellation (Sentinel 1A and 1B) to compare crevasse
locations with those consistent with the von Mises failure criterion.

Following previous work (Nye, 1959; Bindschadler et al., 1996; Veen, 1999; Alley et al., 2018), we first calculate the
principal surface stresses from the ice-velocity time series (see Supplement S2). We then extracted the surface stresses at



crevasse locations identified by the CNN from quarterly velocity measurements between 2015 and 2022. Then, following
Vaughan (1993), we used quarterly principal surface stresses at the location of crevasses to construct failure envelopes and
estimate the tensile strength associated with crevasse formation (see Supplement S2).

## 2.3    A simple geometric model for the fracture properties of firn

Physical models that unify the observations and behavior of near-surface fracture with fracture propagation deeper in the
ice column are important to accurately represent the effects of damage on the effective viscosity of the ice sheet from the
assimilation of the growing observations of crevasse position and crevasse area change (Surawy-Stepney et al., 2023a, b).
Models that describe the effects of porosity on fracture toughness and tensile strength were first developed for metals, ceramics,
polymer foams, and sponges (Ashby and Medalist, 1983; Maiti et al., 1984; Gibson and Ashby, 1999). Though the fracture
toughness of freshwater ice decreases with increasing porosity (Rist et al., 1999, 2002), this relationship remains disconnected
from simulated ice-sheet mechanics, which do not typically represent firn processes. Here, we develop a simple geometric
model for the tensile strength of firn to better understand the apparent disagreement between controlled laboratory experiments
of ice fracture and remote sensing observations of near-surface crevasse location and strain used to derive tensile strengths
from satellite observations.

Our approach uses geometric arguments based on a simple conceptual model for firn pore space to develop a description
for the tensile strength of densifying firn ($\sigma_f$) from the tensile strength of ice ($\sigma_I$) and firn porosity ($P$). We start by assuming
a closed idealized lattice of firn with wall thickness ($t$) and unit cell width ($w$), illustrated in Figure 2. Within this lattice, we
can define the dimensionless ratio of pore space to wall thickness $d = t/w$ where $0 < d < 1$. The porosity can be related to the
volumetric fraction of the firn lattice divided by the volume of the characteristic unit cell, ($w^3$), and described in terms of the
dimensionless ratio of pore space as

$$P = \frac{(w-t)^3}{w^3} = (1-d)^3. \tag{3}$$

We assume the tensile strength of the lattice is reduced by a factor that is proportional to the reduced crack-face surface area
(Fig. 2b) relative to a completely closed lattice (i.e. a lattice representing pure ice). The crack face is a two-dimensional feature
of the lattice that represents the face bisecting the lattice volume that maximizes the area of void space. From the geometry
illustrated in Figure 2, the tensile strength of firn as a function of the dimensionless ratio of pore space can be described by

$$\frac{\sigma_f}{\sigma_I} = \frac{w^2 - (w-t)^2}{w^2} = 1 - (1-d)^2. \tag{4}$$

The idealized relationship between the dimensionless pore space fraction and tensile strength holds in the absence of pore
connections where the density of the firn is very near the density of ice at the depths exceeding the bubble close-off depth,
where firn air trapped in pores can no longer exchange with surrounding voids. To account for pore space connections within
the lattice at lower densities where grains have not yet sintered, we define the relative fraction of pore connections, ($X$), which
we also assume is proportional to the reduced strength across the crack-face:



$$\frac{\sigma_f}{\sigma_I} = X(1-(1-d)^2).\qquad(5)$$

Solving for the dimensionless ratio of the pore space as a function of porosity from Equation 3, we find:

$$d = 1 - P^{1/3},\qquad(6)$$

and thus,

$$\frac{\sigma_f}{\sigma_I} = X(1-(1-d)^2) = X(1-P^{2/3})\qquad(7)$$

Finally, we must determine the relationship between the relative fraction of pore space connections and firn porosity, i.e. $X = f(P)$. Following studies that have used similar geometric arguments to model fracture properties in ceramics and metals (i.e. Jelitto and Schneider, 2018), we assume that the disconnections can be related to porosity by $X = (1-P)^n$, and the tensile strength can be formulated from the geometry of the crack face as

$$\frac{\sigma_f}{\sigma_I} = (1-P)^n \left(1-P^{2/3}\right).\qquad(8)$$

## 3 Results

Our presentation of results begins by examining individual crevasses and confirming their presence in independent optical imagery and radar datasets, we then discuss changes in crevasse patterns across the ASE in the context of other glaciological and oceanographic changes observed over the last 10 years, and the relationship between porosity and tensile strength of firn.

### 3.1 Verifying Crevasse Feature Extent and Constraining Vertical Depth with Optical and Ice-Penetrating Radar Imagery

A comparison of crevasse detections from the SAR data with high-resolution optical imagery and ice-penetrating radar data strongly supports our interpretation that the detected features are crevasses. These datasets interpreted together also reveal that crevasses detected with SAR data are commonly buried so that there is little or no surface expression in optical imagery.

#### 3.1.1 Crevasse Appearance in Optical Imagery

Near the inland onset of crevassing (Fig. 4b), WorldView imagery often directly captures clear crevasses or surface features indicative of crevassing (Fig. 4b), such as regular "swells" likely associated with snow bridges (Supplementary Fig. S3, S4). These features are consistent with features that are also imaged in SAR imagery and ice-penetrating radar data (see Section



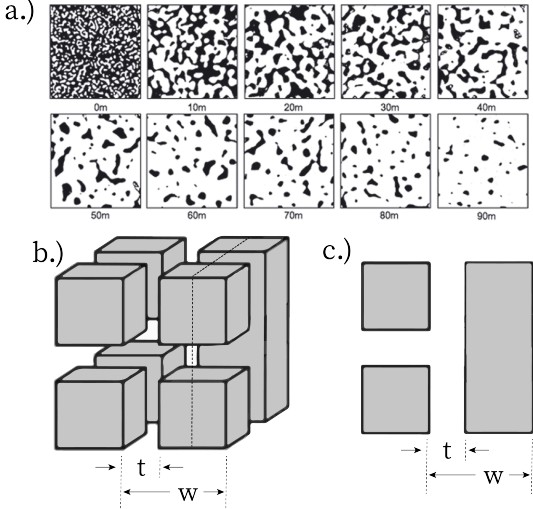

**Figure 2.** Micro CT scan of firn structure revealing the effect of compressed pore space (black) within the firn unit cell (a) shown with the conceptual geometric model of the firn (b) and the crack face used to derive the relationship between tensile strength and porosity (c). The scan was first presented by Lomonaco et al. (2011) using firn cores collected from Summit, Greenland.

3.1.2). In some areas downstream of crevasse onsets, there is extensive crevassing indicated in SAR imagery, but there is no surface expression or ambiguous expression of the crevasses in contemporaneous, co-located WorldView images (i.e. Fig. 4b,

Supplementary Fig. S5c). This suggests that these crevasses are subsurface features – crevasses that have either been buried by accumulation following formation or initially formed below the surface. Examination of spatially coincident ice-penetrating radar profiles confirms that these are subsurface features (Fig. 4e, see Section 3.1.2). This indicates that SAR imagery reveals subsurface crevasses reliably, including occasional near-surface crevasses ($< 10$ m depth).

### 3.1.2 Crevasse Appearance in Ice-Penetrating Radar

There are several commonly observed features recorded in the ultra-wideband ultra-high frequency ice-penetrating radar data that have conventionally been associated with crevasses (Nath and Vaughan, 2003; Williams et al., 2014). Where there is a clear firn/void-space contrast, we see bright reflections from the top of buried crevasses in ice-penetrating radar data (Fig. 5). These reflections appear as hyperbolic scatterers, due to apparent diffraction where the snow bridges re-connect with the continuous firn. When the transmitter is above the crevasse, transmitted wave energy appears to be critically refracted along the

crevasse wall so that little energy is reflected back to the receiver. This results in an apparent low amplitude "shadow" beneath the crevasse void space in the radargram. In the case of basal crevasses observed over the floating ice shelf, there are reflections from the top of the crevasse, where there is an ice/seawater interface. Schematics of the radar wave ray paths and examples of the resulting radargrams that illustrate these characteristic descriptions are shown in Figure 4, Figure 5, and Supplementary Fig. S5-S7.



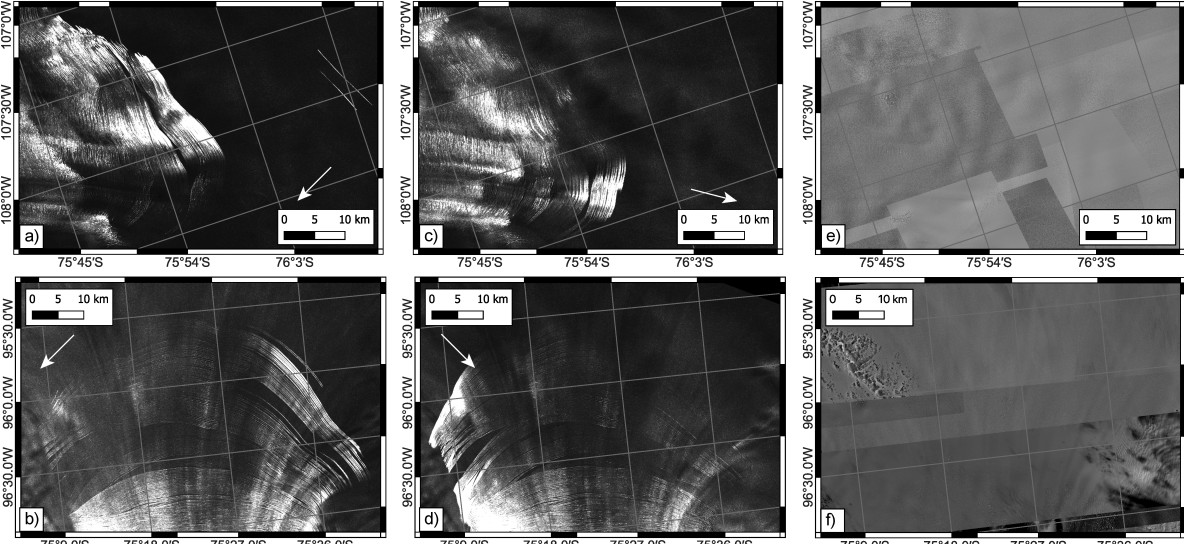

**Figure 3.** Crevasse appearance in ascending (a, b) and descending (c,d) orbits of Sentinel-1A satellite imagery for interior regions of Thwaites and Pine Island Glaciers. Arrows indicate satellite look direction. Also shown is a mosaic of WorldView images (e,f) for the same coverage of Thwaites Glacier and Pine Island Glacier, revealing that many of the crevasses that are readily visible in Sentinel imagery are buried beneath snow bridges that obscure the features in panchromatic optical imagery.

The agreement between crevasses identified in SAR imagery and crevasses visible in the ice-penetrating radar and high-resolution visible imagery gives high confidence that the near-surface crevasses we detect in SAR imagery are robustly identified by the neural network. Observations of the three-dimensional structure revealed in the SAR and ice-penetrating radar data have ramifications for the interpretation of ice tensile strength and the influence of crevassing on ice-sheet structure, which is discussed later in Sections 3.3 and 2.3. We next evaluate changes in the crevasse area recorded in the SAR imagery time series and their implications for understanding the material properties and fracture mechanics of firn/ice.

## 3.2 Crevasse Expansion in the Amundsen Sea Embayment

Results from the algorithm applied to the SAR image time series are summarized in Figures 6 and 7. We identify three qualitative changes in crevasse area and spatial density that together result in an overall increase in the damaged surface area of the Amundsen Sea Embayment:

- Crevasse initiation in previously uncrevassed ice that is farther inland than any previously observed crevasse.

- The advection of crevasses from upstream into undamaged areas downstream.

- New crevasse appearances that overprint preexisting fractures that have been advected from upstream. These changes are discernible in the spatial density of crevasses and the patterns of crevasse orientations.



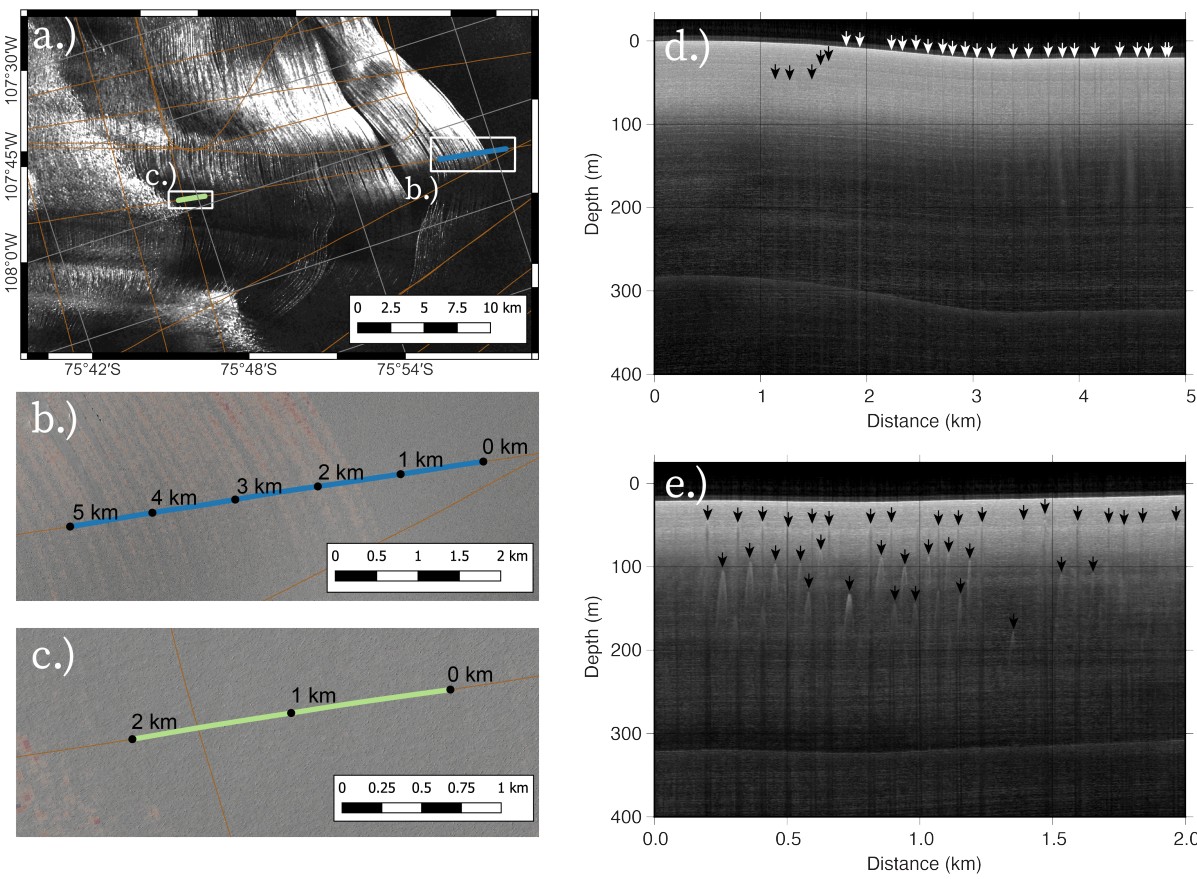

**Figure 4.** (a) SAR backscatter intensity image with flight lines of ice-penetrating radar that cross the (b) onset of surface crevassing detected in U-net (red) with panchromatic optical imagery and (c) buried U-net detected crevasses (red) downstream shown with panchromatic optical imagery. Buried crevasses are visible in radar profiles (d and e). Subpanels (d) and (e) correspond to radar profiles along flight lines shown in (b) and (c), respectively. White arrows indicate crevasses that are expressed at the surface while black arrows indicate subsurface crevasses with hyperbolic diffractors. The vertical position of the black arrows indicates the depth that crevasses first appear in radar data relative to the surface.





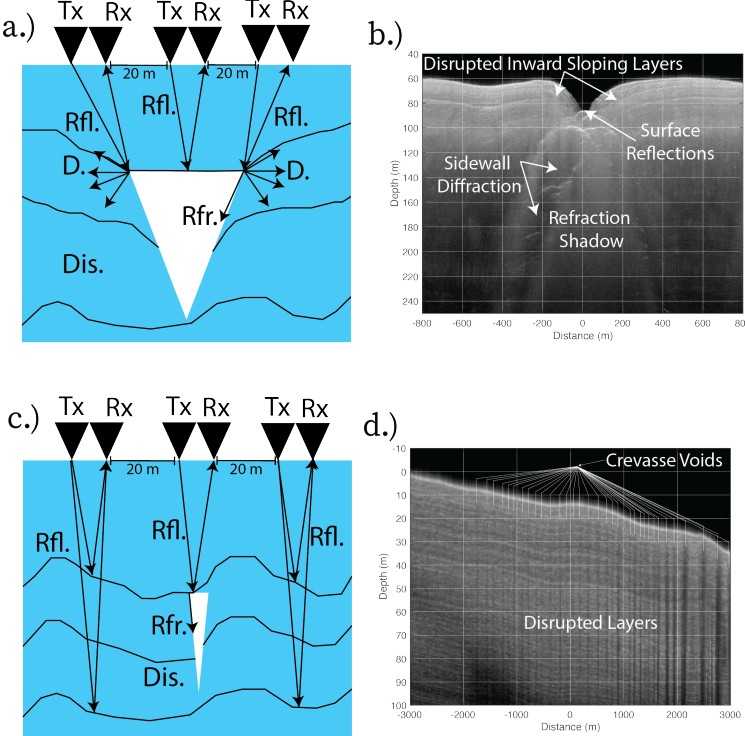

**Figure 5.** transmit and receive paths for different crevasse geometries. The transmitter (Tx) and receiver (Rx) are shown with illustrations of the transmit wave path. Features in shown in radargrams are labeled on the wave paths: hyperbolic reflections from the crevasse top (Rfl.), refraction along the crevasse wall (Rfr.), hyperbolic sidewall diffraction (D.), and discontinuous subglacial layering (Dis). Also shown are (a-b) the reflections we attribute to narrow crevasses we observe in the interior near-surface and (c-d) larger near-surface crevasses that have opened on the ice shelf.

The broad-scale crevassing patterns differ between the major glacier systems we examine in this study and are correlated with changes in ice velocity and effective stress expressed in the glacier surface velocities (Fig. 7). From 2017–2022, Pine Island Glacier sped up, accelerating from the grounding zone (∼500 m/yr) to the deep interior. As the glacier accelerated, likely due to changes in the buttressing capacity of the ice shelf due to a large iceberg calving event in 2017, the crevassed area increased (Fig. 7b; Joughin et al., 2021). The crevassed area then increased more slowly from 2018 to 2020 remaining almost constant. This stagnation in the crevassed area was followed by a second rapid increase in crevassed area from 2020–2021, which is contemporaneous with another major calving event. Thwaites Glacier also accelerated between 2017-2022, speeding up by 400 m/yr near the grounding zone (Benn et al., 2022). This speedup has been attributed to the development of full-depth fractures near a prominent pinning point on the Thwaites Eastern Ice Shelf (Benn et al., 2022), and also coincides with enhanced sub-ice-shelf melting in the ASE (Davis et al., 2023). During this time period, the crevassed area of Thwaites Glacier increased by 10% and expanded inland of the glacier grounding line; crevassed area density also increased in previously crevassed regions. The glaciers that feed Dotson and Crosson Ice Shelves did not systematically speed up during this time period and





**Figure 6.** SAR time series of inland crevasse advection and nucleation between 2018-2019 in the upper reaches of the crevassed trunk of Thwaites Glacier. The U-net crevasse position of a scene shot in March 2018 (cyan) overlain with subsequent crevasse positions (orange). The crevasse position images are transparent so that where the locations overlap the image appears green. Arrows highlight different mechanisms of crevasse area change: blue, crevasse initiation in previously uncrevassed areas; orange, advection into previously uncrevassed areas; and purple, new crevassing in areas with preexisting fractures.



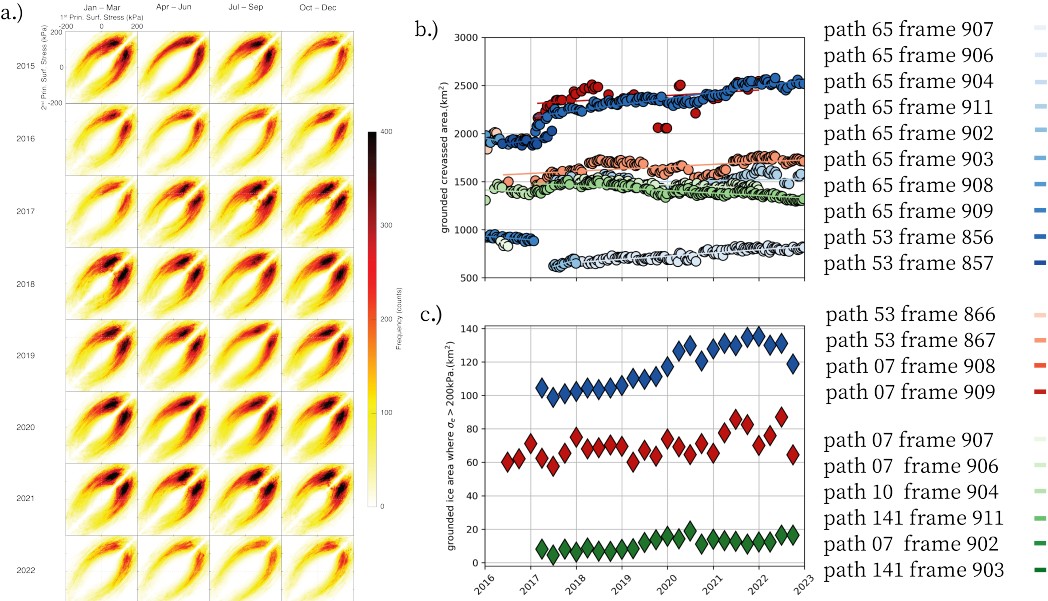

**Figure 7.** (a) Principal surface stress failure envelopes of Thwaites Glacier crevasse time series from 2015–2022, following Vaughan (1993). (b) Crevassed area for the grounded regions of Pine Island (blues), Thwaites, (reds), and the glaciers feeding Dotson and Crosson Ice Shelves (greens), with (c) the area where the effective stress (calculated using the surface membrane stresses) exceeds 200kPa.

crevasse areas here remained steady except for a sutble increase in crevassed area in 2017. These observations indicate that the expansion of crevasses is likely due to increases in surface stresses associated with higher surface strain rates.

### 3.3 Failure envelopes and a data-driven tensile strength model of near-surface fractures

Following Vaughan (1993), maps of surface stress and fracture location were used to create ovaloid failure surfaces (principal surface stress plots) for each quarterly velocity product. These failure envelopes do not change substantially over time (Fig. 7a). This ovaloid relationship between principal surface stresses and brittle failure matches the predictions of the von Mises failure criterion (Vaughan, 1993). Comparing surface stress values between crevassed and uncrevassed ice suggests that the tensile strength of the near-surface is bounded (Fig. 8). For Thwaites Glacier (other glaciers in the ASE have similar results), this approach constrains a failure envelope consistent with a tensile strength between 75 and 210 kPa. These estimates of the tensile strength of the ice sheet near surface agree with limits cited elsewhere in the remote-sensing literature but disagree with failure limits determined from laboratory measurements conducted on pure ice ($\sim 1800$ kPa; Haynes, 1979).



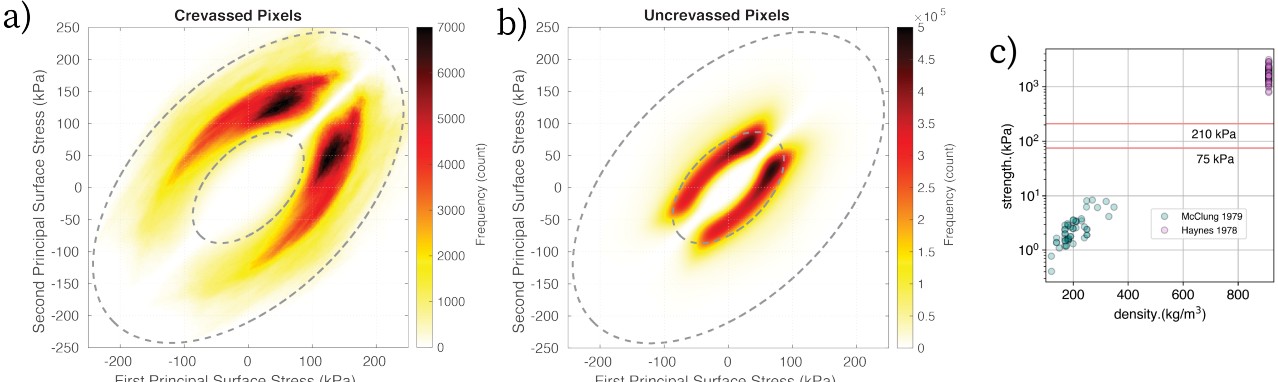

**Figure 8.** Principal surface stress failure envelopes observed on Thwaites Glacier are shown for (a) crevassed and (b) uncrevassed regions of the glacier identified with U-net. These evelopes bound the tensile strength of the surface (dotted lines). Also shown are the (c) experimental tensile strengths inferred from laboratory experiments conducted on snow and ice. The tensile strengths inferred from the surface observations in this study are also shown (red lines).

## 3.4 Geometric Model

Our geometric model for the tensile strength of firn suggests that porous snow is much easier to fracture than ice. Figure 9 shows a comparison of the tensile strength model fit with tensile strengths determined from snow experiments and experiments on ice conducted over a range of ice temperatures and grain sizes (McClung, 1978; Haynes, 1979). The model shows encouraging agreement with lab experiments and suggests that the near-surface fractures we observe in the Amundsen Sea are consistent with the fracture mechanics of polar firn.

## 4 Discussion

### 4.1 Drivers of Inland Crevasse Expansion

Inland migration of crevasses appears to be linked to ongoing inland acceleration and inland effective stress changes in response to mass loss near the grounding zone. Pine Island, Thwaites, and the neighboring Haynes, Pope, and Kohler Glaciers in the ASE are losing mass at a greater rate than any other glaciated region in Antarctica (Shepherd et al., 2019; Rignot et al., 2019). Currently, submarine melt appears to pace mass loss and ASE glacier retreat (Jacobs et al., 1996; Jenkins et al., 2010, 2018), but changes in the buttressing capacity of ASE glacier ice shelves due to calving have also been linked to ephemeral inland acceleration (Joughin et al., 2021), and loss of ice-shelf buttressing has been hypothesized to be a major driver of retreat in a future warmer climate (DeConto and Pollard, 2016; Alley et al., 2023). Glaciers in the ASE accelerate to balance changes in driving stress, however, the strength and roughness of the bed isn't uniform, and thus some areas of Thwaites and Pine Island in particular have steepened while other areas have become more flat. The heterogeneous pattern of thinning we observe across





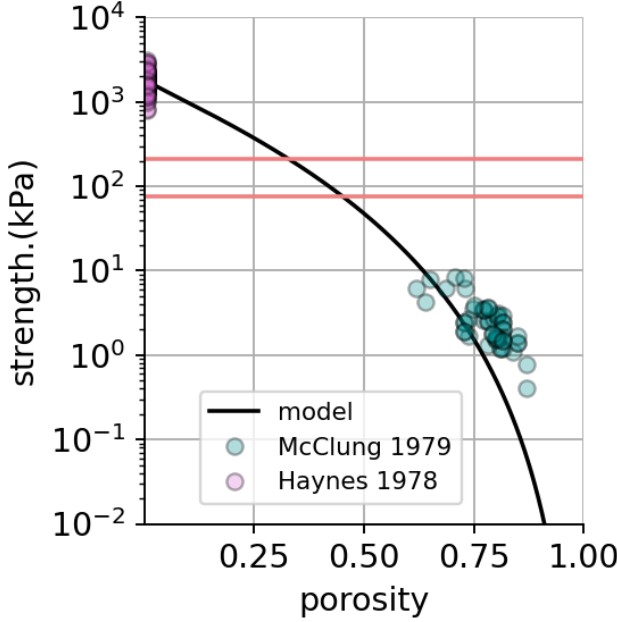

**Figure 9.** Model fit for tensile strength of ice and firn as a function of porosity. Data used to fit the model are shown from studies on ice (Haynes, 1979) and porous snow (McClung, 1978). The best-fit model would suggest an exponent, $n = 3.6$, and tensile strength, $\sigma_I = 1800$kPa. The range of tensile strengths measured in laboratory experiments for solid ice (i.e. Haynes, 1979) demonstrate the dependence of fracture on temperature and physical properties of crystal grains.

Thwaites reveals the complexity of this relationship (Supplement Videos, Hoffman et al., 2020) and agrees with patterns we
see in the effective stress field.

This also appears to explain why crevasses jump inland, sometimes skipping several kilometers of uncrevassed area to initiate upstream in regions where the surface stresses are higher. In these areas, surface slopes are steep often on the lee side of local surface elevation highs associated with large-scale ridges in subglacial topography and locally enhanced basal resistance (Fig. 10; Hoffman et al., 2022). In this way, surface crevasses appear to be connected to glacier sliding as the relationship between
basal shear stress and glacier sliding velocity mediates inland stress transmission.

## 4.2   Depiction of Crevasses in Ice-Penetrating Radar Data

Joint inspection of co-located SAR-detected crevasses and ice-penetrating radar profiles revealed that crevasses identified in the SAR imagery are also expressed in ice-penetrating radar data. Because ice-penetrating radar data image the depth of englacial structures, these data can add context to the crevasses we identify in satellite SAR imagery. In the discussion that follows, we
begin by describing the manifestation of crevasses in the ice-penetrating radar data. We then consider the implications of the features we image for crevasse geometry and depth.





**Figure 10.** Crevasse observed in Sentinel-1A SAR imagery on May 28, 2019 (a) and June 9, 2019 (b), highlighting a major crevassing event. Crevasse probability index from this study and accumulation radar data collected by the International Thwaites Glacier Collaboration airborne geophysics campaign (magenta profile in the top panels) are also plotted (c). Also included (d) is a profile of ice-penetrating radar data collected as part of the Aerogeophysical Survey of the Amundsen Sea Embayment, Antarctica (AGASEA) project (cyan profile in the top panels). Contours in the top panels are bed elevation from BedMachine Antarctica.



Most crevasses we observe in the survey are imaged either as voids that start at the surface or bright hyperbolic reflectors in the near-surface ($< 100$ m depth) with voids beneath the hyperbolas that interrupt layering at depth (Fig. 5c, Supplement Fig. S3, S4). Occasionally, we observe hyperbolic reflections deeper in the column – sometimes $100 - 150$ m below the surface. These depths are below the observed firn-ice transition measured from ASE firn cores (Gow et al., 2004). These features could be crevasses we image off-nadir that appear deeper in radar profiles than they are in reality. They could also reflect larger flaws in the ice (>4m) that have been shown using linear elastic fracture mechanics to reduce the tensile strength of ice (i.e. 59 kPa Lai et al., 2020). In all cases, the radar does not image hyperbolic reflectors deeper than $\sim 200$ m except where basal crevasses on ice shelves are also present. Voids and hyperbolic reflectors often begin below intact near-surface layering. Features we identify as buried crevasses only appear downstream of onset zones in areas where the surface expression of crevasses is muted in SAR and optical imagery (i.e. Figure 4c). Except for several very wide crevasses on the ice shelf, englacial layering in the vicinity of the crevasses does not noticeably slope into the crevasse.

The bright reflectors we observe buried beneath the surface are consistent with hyperbolic scatterers due to apparent diffraction where snow bridges re-connect with the continuous firn. Beneath these hyperbolic scatterers, very little energy is returned to the receiver. Similarly, where crevasses penetrate to the surface in optical and SAR imagery, ice-penetrating radar data show a significant decrease in energy reflected back to the receiver. This low-energy void appears to be caused by the steep angle of crevasse side walls that facilitate the refraction of energy along the wall away from the radar receiver. This results in a low-energy void beneath the top of the crevasse to depths below the bed reflector in neighboring traces to the noise floor of the receiver. The voids clearly cannot be interpreted as a proxy for crevasse depth, their appearance does change though with the width of the crevasse feature. Near crevasse onset zones where crevasses occur at the surface, sidewall hyperbolic reflections, if present, are very weak. Further downstream, where crevasses are wider (Supplementary Fig. S7-S8), all features discussed above are more notable (i.e. Figure 5c compared to Figure 5e). For these larger crevasses, we also observe internal layers dipping towards the crevasse, which we do not see in the vicinity of narrower crevasses we image upstream.

Crevasse depths remain extremely difficult to measure directly. With ground-based and airborne radar technology, it remains possible only to observe how the features affect local stratigraphy and flow expressed in nearby layer slopes. Because energy is critically refracted away from the receiver below the exposed surface crevasses (or the initial firn-air reflection of buried crevasses), we never image the base of the crevasses we identify in SAR imagery and ice-penetrating radar data. The lack of diffraction hyperbola from crevasse sidewalls and the width of the void space suggest that the near-surface expression of the features is small relative to the trace spacing of the radar system; however, narrow crevasses have still been observed to penetrate deep into the near-surface (Lindner et al., 2019). The most compelling evidence that suggests the buried crevasses we image with radar do not penetrate below the firn is the absence of hyperbolic scatters below depths of 200 m and the limited impact the surface features appear to have on nearby ice flow.

In areas of crevasse onset, where englacial stratigraphy below 200 m depth is present on either side of the void space, the stratigraphy remains continuous and layers do not dip into the low-energy void. This suggests that these crevasses minimally disrupt the flow, which tends to close large crevasses at depths where the overburden stress of the ice and firn exceeds the horizontal tensile stress of the ice column ($\sim 100$ m depth van der Veen, 1998). We see this effect downstream where the




expression of the crevasse void space and sidewall reflectors change with increasing horizontal strain rates. After crevasses
open, the confining overburden pressure of firn and ice above drives viscous flow into the void space. This can cause the draw-
down of englacial layers on either side of the crevasse, which we use as a diagnostic tool to constrain the depth of crevasses
(Fig. 5 and Supplementary Fig. S8). Over the ice shelf, the width of the low power return beneath apparent surface crevasses
increases, and sidewall reflections of near-surface crevasses become visible at intermediate depths (Fig. 5). Near these wider
crevasses, layers steepen, sloping into the apparent void at intermediate depths ($\sim 40$ m). This suggests that the features are
large enough and extend deep enough that they can affect ice flow well below the surface. Over the ice shelf, we also image the
tips of basal crevasses that extend upward $400 - 500$ m from the ice-shelf base (Supplementary figures S8, S9). Englacial layer
slopes change above these basal crevasses, suggesting again that these much larger features affect flow in ways the interior
surface crevasses do not. These basal crevasse features extend three times deeper into the ice shelf than the deepest hyperbolic
defractors associated with surface crevasses and reveal how near full-thickness crevasse features appear in the same radar data.

### 4.3 Firn tensile strength

In the top half of the ice column, where viscosity changes are small, we assume that the tensile stresses are nearly uniform. As
a result, crevassing in this region can define a strength threshold and suggests that the densifying snow and firn are weaker than
the solid ice below.

Below the near-surface, where crevasses have opened, compressive overburden pressure acts to close open-air crevasses
(Weertman, 1973; Smith, 1976, 1978; van der Veen, 1998). Our observations are consistent with crevasses closing below 200
m depth and suggest that compressive overburden stresses in combination with firn strengthening with density and depth, arrest
dry firn crevasse propagation.

In the literature, inconsistent attention to the strength of firn has led to reported disagreement between the fracture properties
of ice derived from laboratory measurements and estimates from in situ and remote-sensing observations. Tensile strengths
measured in laboratory experiments are significantly higher than those determined from remote sensing observations. The
values we observe (75 to 210 kPa) are within ranges reported from other satellite remote-sensing observations of crevassing
in Antarctica (Vaughan, 1993) and Greenland (Grinsted et al., 2023). These ranges are consistent with our geometric model
which predicts firn fracture where effective stresses exceed $75 kPa$ when the bulk average porosity is greater than $0.4$. This
suggests that the crevasses and associated tensile strengths determined from remote sensing of surface strain rates likely reflect
near-surface fracture in firn, where the fracture toughness and tensile strength are reduced by the porosity of the firn (Smith
et al., 1990; Fischer et al., 1995; Rist et al., 1999, 2002).

### 4.4 Implications of firn crevasses


Our observations of inland crevasse migration on Thwaites and Pine Island Glaciers are consistent with a porosity-dependent
fracture model that predicts firn fracture preceding crevasse penetration through ice. Our model suggests that surface stresses
in the ASE open surface crevasses that penetrate through the porous firn. We estimate that the firn in the ASE has a tensile
strength between 75 and 210 kPa consistent with a depth-averaged porosity between 0.25 and 0.5 ($\sim 450 - 750$ kg/m$^3$ firn





density). These crevasses would not be expected to penetrate deeper into the ice column. These results have implications for understanding the current impact of fractures on ice dynamics in the ASE and the vulnerability of the ice sheet to fracture-related collapse and ice shelf weakening. They also carry implications for determining the mass balance of crevassed sectors of the ice sheet. We devote the remaining discussion to these two related topics and highlight some directions for future observational work.

### 4.4.1   MICI instability

Ovoid failure envelope analysis and porosity-dependent tensile strength modeling suggest that tensile strength estimates for ice that use values calculated from observations of surface fractures underestimate the bulk strength of the ice sheet (i.e. the tensile strength of solid ice is greater than 210 kPa).

     Models of cliff calving that underpin the MICI hypothesis often treat the ice column as a homogeneous material (Parizek

et al., 2019; Clerc et al., 2019), with calving thresholds that are sensitive to the prescribed material strength (i.e. DeConto and Pollard, 2016). Our work shows that the strength of the ice column is likely heterogeneous with depth, and observed surface crevassing would not occur without a porous (and therefore) near-surface firn layer. Remote sensing estimates of tensile strength are more representative of surface properties, and therefore should be treated as a lower limit on material strength, as they would yield unrealistically high rates of full-thickness fracture in MICI models.

### 4.4.2   Ice-shelf weakening

Surface crevasses have also been suggested to signal ice-shelf weakening (i.e. Lai et al., 2020; Alley et al., 2021; Zhao et al., 2022; Surawy-Stepney et al., 2023a) with several studies recently highlighting the potential impact of ice-shelf surface crevasses on ice flow (Lai et al., 2020; Zhao et al., 2022; Surawy-Stepney et al., 2023a). Our ice-penetrating radar observations of near-surface crevasses inland and across the grounding zone suggest that near-surface crevasses do not significantly change

character as they advect onto the ice shelf and that many of these features are still confined to the near surface (Supplementary figures S6, S7). The depth of these features may explain the weak relationship found by Gerli et al. (2023) between ice-shelf crevasses identified by Izeboud and Lhermitte (2023) and inferences of ice-shelf fluidity derived from contemporaneous surface velocity and ice thickness observations. Our results in connection with the findings of Gerli et al. (2023) suggest that surface crevasses expressed in satellite remote sensing datasets (e.g. Lai et al., 2020; Zhao et al., 2022; Surawy-Stepney et al., 2023a;

Izeboud and Lhermitte, 2023) only weakly affect the bulk viscosity of ice shelves. In contrast, some of the basal crevasse features we observe with radar penetrate through almost $75\%$ of the ice-shelf column ($> 600$ m), $3\times$ the depth of the deepest features associated with buried surface fractures in radar data (supplementary figures S6, S7). These basal crevasse observations should be assimilated into ice shelf models as these features have been observed to affect ice-shelf flow (i.e. Jeong et al., 2016).





### 4.4.3 Firn air content

Our observations of near-surface crevasse opening in the firn also have implications for satellite remote-sensing observations used to calculate glacier mass loss. Firn depth and density estimates are required to convert altimetry observed height changes to mass changes and remain a major source of uncertainty in mass balance studies (i.e. Morris and Wingham, 2015; Verjans et al., 2021). Firn depth and density estimates are also required to convert firn/ice column thickness and velocity observations to discharge fluxes across the ice-sheet grounding zone (Rignot et al., 2019). Both of these methods rely on accurate descriptions of the firn air content, which can change with horizontal strain rates.

Horizontal stresses can affect firn air content via three different mechanisms. Horizontal divergence can stretch the firn, effectively thinning the firn column as suggested by Horlings et al. (2021). Horizontal stresses can also enhance creep-driven densification (Oraschewski and Grinsted, 2022), and as we've shown in this study, horizontal stresses can fracture the firn, opening large volumes of void space in the near surface.

In grounded ice regions, elevation change observations of the ice-sheet surface can be corrected for changes in firn air content, and glacial isostatic adjustment to estimate grounded ice-sheet mass change. Because this is a measurement of vertical height change, the effect of horizontal void space introduced by crevasses opening in the near-surface affects the estimate of ice-sheet thinning or thickening below based on how the effects of horizontal divergence-induced layer thinning and creep-enhanced densification are parameterized in the treatment of firn air content. A correction that separates enhanced creep-driven firn densification (Oraschewski and Grinsted, 2022) and horizontal divergence induced stretching (Horlings et al., 2021) from horizontal divergence associated with crevasse opening could account for the near-surface changes in firn densification. This effect would tend to decrease the enhanced densification rates calculated by (Horlings et al., 2021) that assume all horizontal divergence in the near-surface is accommodated by vertical densification.

Surface crevasses also affect estimates of ice flux across the grounding zone. Because this measurement is made with respect to a vertical gate at the grounding zone where the ice goes afloat, horizontal and vertical variations in firn air content affect calculations of discharge flux. In regions where the near-surface crevassed area observed with SAR imagery has increased and presently accounts for $\sim 10 - 40\%$ of the total surface area, such as lower Thwaites Glacier, the void space associated with open crevasses contributes substantially to the total firn air content. Assuming the depth of the void space is on average $\sim 20$ m, the area occupied by crevasses may represent between $2 - 8$ m of unaccounted air content in the near-surface that would affect the discharge calculations of Thwaites Glacier made from horizontal velocity and surface thinning observations. These approach the uncertainty in bed topography measured with ice-penetrating radar crossover discrepancies in ice thickness (mean 8m, RMS 47m; Holt et al., 2006) and can change over time with the stress state of ice near the grounding zone. Together, the firn air content (consistent with the horizontal strain in areas that remain uncrevassed) and the crevasse void space should both be included in flux-gate analysis of glacier systems.



## 5 Conclusions

We present new observations of changes in the fracture behavior of the ice-sheet near-surface of Thwaites Glacier and its neighbors in the Amundsen Sea Embayment. Our results indicate that changes in the crevassed area are generally correlated with increases in surface stresses in response to ongoing ocean-driven acceleration. Although our observations are consistent

with observations of ice-sheet near-surface tensile strength ($\sim 70-200$ kPa) determined from remote sensing, these failure limits are lower than the failure limits predicted by laboratory experiments conducted on solid ice. We have shown that these inconsistencies can be resolved by accounting for the effects of firn porosity on near-surface tensile strength. Using this new model, we showed that the increase in the crevassed area on Thwaites Glacier is consistent with the increase in the area where effective surface stresses exceeding a critical tensile strength for the near-surface firn. At present, the near-surface crevasse

features appear to be a symptom rather than a driver of acceleration and retreat in the ASE. The void space introduced by crevasses in the near-surface may not have a large effect on full-thickness failure rates but likely does affect calculations of firn air content in rapidly accelerating and crevassed areas of the ice sheet. The time series of crevassed-area evolution that we use to support these results presents a valuable target for models that incorporate near-surface fracture or continuum damage mechanics. Finally, the ongoing collection of SAR images can also be processed rapidly enough using our automated

framework that live crevasse detection in areas where researchers are conducting fieldwork is possible.

*Code and data availability.* The fracture fields for Amundsen Sea glaciers are available at the NSIDC. For live maps of crevasse patterns in new locations, reach out to aoh2111@columbia.edu. The scripts for creating the figures and crevasse propagation rate data products shown in this study are available at the GitHub repository: https://github.com/hoffmaao/thwaites_crevasses. The masking software used to build the georeferenced masks is available at https://github.com/hoffmaao/EarthMasker.

*Video supplement.* Videos of crevasse migration on Thwaites Glacier are available at https://doi.org/10.5446/62770, and https://doi.org/10.5446/62771.

*Team list.* The complete member list of the GHOST science team can be found at https://thwaitesglacier.org/projects/ghost.

*Author contributions.* AOH and KC theorized the study. AOH wrote the software to segment images and implemented the neural network crevasse detection method following the work of CYL. AOH, KC, and NDH interpreted the radar grams collected by operation ice bridge.

IJ processed and created the velocity time series used to calculate surface strain. AOH wrote the first draft of the manuscript. All authors contributed to the editing of the manuscript.



*Competing interests.* The authors declare no conflicts of interest.

*Acknowledgements.* The work was supported by a NASA FINESST award (grant 80NSSC20K1627), the NASA sea-level change team (grant 80NSSC17K0698), and the NSF-NERC International Thwaites Glacier Collaboration (grant OPP-1738934).



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
