# Peer review of "Inland migration of near-surface crevasses in the Amundsen Sea Sector, West Antarctica"

_EGUsphere, 2023_

## Referee Comment (RC2)

**Review of "Inland migration of near-surface crevasses in the Amundsen Sea Sector, West Antarctica"**

In this study the authors develop a method of mapping crevasses on the grounded ice of the Amundsen Sea Sector of West Antarctica and use airborne radar data to assess the depth of the crevasses they identify. The crevasses are shown to be in the upper part of the ice - largely confined to the firn. They also report changes in crevassing on the grounded ice across the Amundsen Sea Sector and suggest this is linked to changes in the dynamic behaviour of the region's ice streams since 2015. They go on to investigate the tensile strength of the ice surface (using a von Mises stress-criterion model) using their crevasse data and satellite-derived ice velocity data, and find strengths compatible with porous firn - estimated using a simple model fit to data. Overall, I think there are many good things about this paper: what we can see of the crevasse maps look good, the methodology for generating them is explained well and the data is validated; the conclusion that the crevasses on Thwaites rarely penetrate deeper than the firn is convincing and timely, given the current discussion regarding the impact such crevasses could have on ice flow. Similarly, the conclusion that surface fracture occurs according to a von Mises-like criterion is promising.

However, some of the ideas in the article are not developed fully or demonstrated as convincingly as they could be. It is possible that some of this is to do with the presentation of results, which should be improved throughout. I think the paper on the whole might try and do a little too much. For example, it doesn't benefit from the inclusion of the firn model. Hence, I believe it should be published, and will be useful to the community, but requires a number of revisions before publishing.

**Changes in crevassing:**

In general, some more work is required to demonstrate that the result in the title is widespread. This could be improved by showing a greater number of crevasse maps, rather than focusing on a small portion of Thwaites Glacier (as in Figure 6) and some large scale statistics like crevassed surface area (as in figure 7b-c).

The inland propagation of crevasses is shown in the main paper only in figure 6, where we can see some evolution on the south-eastern side of the crevasse field between March and June 2019. However, the larger region in Figure 1b (where the colours are labelled the other way round, or perhaps incorrectly?) shows a more varied picture with some locations showing that crevasses have been advected downstream and not replaced, and others where crevasses have developed further upstream.

Figure 7c shows that the crevassed area on Pine Island Glacier (imaged by path:53, frame:857) increased by 25% between 2016 and 2023. This is a big change so it would be great to see some maps showing the difference in crevasse area at the start and end of the timeseries, especially as there is no indication of uncertainty in the plots.

There is quite a lot of talk about correlation of increased crevasse area with changes in speed or increased surface strain rates. However, we are not presented with any data regarding changes in speed or surface strain rates in the ASE and no analysis linking this

to changes in crevassing has been shown.

**Firn model:**

I found section 2.3 a bit difficult to read, and the diagram in Fig. 2 is not particularly illuminating. However, the idea is pretty simple and I think can be reduced to something like the following argument:

The porosity $P$ of the firn is defined as the ratio of the volume of air contained in the pores to the total volume. The fractional area of air intersected by a plane is then $\sim P^{2/3}$. Hence, the tensile strength of the firn, which is proportional to the area of ice intersected by a plane normal to the background tensile stress, is reduced by a factor $\sim (1 - P^{2/3})$ compared to firn of zero porosity. At high porosities, some of the filaments of ice separating pores from each other break and the pores can merge. To account for this, we follow Jelitto and Schnewider (2018), and introduce an additional factor $(1-P)^n$ where n is a parameter we fit to data from laboratory experiments. Hence, we find:

$$\frac{\sigma_f}{\sigma_I} = (1 - P)^n \times (1 - P^{2/3}). \tag{1}$$

This model does not constitute a major novelty of the article (as it is described in Jelitto and Schnewider (2018) and is not validated by tensile strength estimates later) so this argument could be moved into the discussion where it could sit alongside figure 9.

However, I am not sure the description of the firn model and figure 9 add much to the article, so would just suggest removing them. The porosity of the firn is not known (at least, it is not shown in the article) so what figure 9 serves to do is show that the tensile strengths found from the estimates of surface stress are not inconsistent with firn described by such a model. This is fine except that these data are compatible with pretty much any model where tensile strength decreases monotonically with porosity (especially models with two fitting parameters). The fact that we do not know the porosity of the firn means that the strengths calculated from the satellite data do not provide evidence for the model. Furthermore, the model itself does not provide any further evidence for the crevasses initiating in firn than figure 8c plus the statement that firn tensile strength should decrease with porosity. Overall, in my opinion, there is not really a need for this in the article so I would recommend taking these sections out. I think figure 8c is useful and interesting, and does the job at demonstrating that the crevasses are likely to initiate in firn. This might have the added benefit of improving the overall flow of the article.

**Von Mises stress criterion:**

The ellipses shown in figures 7 and 8 look great. It is evident that the vast majority of the crevassed locations on Thwaites sit inside the bounds of the yield region shown, and the vast majority of the un-crevassed locations sit at effective stresses lower than the inner yield curve. There are a couple of things I hope the authors can elaborate on.

Firstly, the ellipses show that reaching a stress threshold between 70 and 200 kPa is a necessary condition for fracturing. However, I wonder whether the authors could comment on whether it is a sufficient condition? The panels 8a and 8b have different scales on the colourbars (necessary because of the huge class imbalance between crevassed and un-crevassed locations), so it is not clear whether a measured surface stress between 70 and 200 kPa is a good indicator of whether there will be a crevasse. Perhaps the authors could show a map of where stresses in the Amundsen Sea Sector exceed 200 kPa. The figures at the moment do not rule out the possibility that the von Mises stress criterion is a good way of determining where crevasses are not likely to be, but might not be particularly skillful at determining where crevasses are likely to be (potentially because the strength of the ice varies too much due to differences in firn porosity?). If this turns out to be true, it is an interesting point of discussion.

It is also not clear exactly which area of Thwaites Glacier is being used to test the stress envelopes. Is it the entire crevassed area on Thwaites or the small section at the upstream edge of the crevasse field? It would be great to see this analysis extended to the rest of the Amundsen Sea Sector (we are told other glaciers show similar results). If the concern is that the porosity over the area has to be relatively similar to get a well-defined stress envelope, then the area could be broken up into smaller sub-regions. Given the potential here it seems a shame to limit the conclusions to a particular glacier rather than saying something more general about the failure behaviour of ice.

**Specific comments**

1. I would use the word "ellispe" or "elliptic" throughout the article when describing the von Mises stress envelopes, rather than "ovoid/ovaloid" unless that's standard.

2. Figure 1. I think the colours for panels b and c might be labelled the wrong way round. If not, it would be better if they were consistent with figure 6.

3. Line 90: It is not clear to me why the use of Sentinel-1B images would result in geometric inconsistency.

4. Line 104: Is there any scaling applied when the images are converted to 8-bit integers, for example by something like requiring the mean backscatter to correspond to a value of 100. Otherwise, do the values concentrate at 0?

5. Line 127: I would write "Adam" optimisation rather than "Adams" optimisation.

6. Line 137: This part about how the images were combined - including the statement about the translational equivariance - is a little unclear to me. Is it the case that the padding is applied to an image patch prior to its processing by the neural network, and that the processed image could be in a different location?

7. Line 237: "interpretted" should be "interpreted'

8. Line 244: Figure 4b seems to be referenced both as evidence for the visibility and invisibility of crevasse features in optical imagery. Perhaps this should be 4c?

9. Line 276: Could you provide a citation for the 500 $ma^{-1}$ increase in speed deep into the interior of the Pine Island Glacier betwen 2017 and 2022?

10. Figure 6 is very nice, but it is not really discussed in the main text. Perhaps some details specific to the figure should be discussed in section 3.2.

11. I think Figures 6-8 could be restructured to make them more compelling. For example, it would be great to see a before and after image for Pine Island, Thwaites and Pope/Smith/Kholer along the lines of figure 6 to accompany the timeseries of crevasse area change in Figure 7. It is a bit of a pain to go back to figure 1 to see the path/frame numbers when looking at figure 7b/c, so would it be possible to reproduce the acquisition footprints here as well? I think it would be better to either make 7a its own figure or combine it with figure 8 to improve the focus of this part of the article. As it stands, these sections seem to jump about a bit and lack focus to some extent.

12. Figure 7a: Could the authors comment on why the density of pixels seems to change from year to year at the same location? For example, the densities seems uniformly lower on Thwaites in 2022 than 2021. The corresponding crevasse area timeseries in 7b show that the total number of datapoints should be roughly the same so the datapoints are more widely distributed? What causes this?

13. Figure 7a: There are some faint dashed lines in the figure which, in the top row at least, indicate the coordinate axes $\sigma_{1,2}$. These seem to change position in years 2017-2020. Given the statement on line 290 I guess this is a mistake?

14. Line 292: Is this calculated as an average over the whole timeseries of crevasse maps and velocity fields or does it correspond to a specific date? If the latter, why was it chosen?

15. Line 294 and Figure 8: Can the authors elaborate on how the dashed curves were calculated as these form the basis for the observed strengths. For example, are they the closest ellipses to a contour on to the density plots (10% of the maximum density defined over some area could be an option)? There is also some inconsistency between how the bounds are reported in the article - sometimes as $75 - 210$ kPa and other times as $70 - 200$ kPa.

16. Figure 9: Could you include in the caption that the red lines are the same as those in the previous figure?

17. Section 4.4.2: This section is not quite right. The maps of fractures in the studies referenced deal in large part with the surface expressions of ice shelf rifts and basal crevasses, not with the kinds of crevasses considered in this article. It is true that the surface crevasses seen here, when advected onto the ice shelves do not change a great deal but this has little relevance to the other studies referenced here, or to the work of Gerli et al., 2023. I would recommend removing this section and adding a sentence to the end of the first paragraph at the start of section 4.4 explaining that your results show crevasses of this type have no relevance to ice shelf weakening.

18. Line 480: "surface strain" should be "surface strain rate".

**References:**

Jelitto, H. and Schneider, G.A., 2018. A geometric model for the fracture toughness of porous materials. Acta materialia, 151, pp.443-453. https://doi.org/10.1016/j.actamat.2018.03.018

---

## Author Comment (AC1)

Author response for *Inland migration of near-surface crevasses in the Amundsen Sea Sector, West Antarctica*

*We thank the reviewers and the editor for their thoughtful comments that help improve our manuscript. We've provided a line by line response to comments by the reviews here. The reviewers' comments are in bold; our responses are in italics.*

**This paper adopts a set of methods to map grounded ice crevasses in the Amundsen Sea Embayment. The analysis shows the presence of surface crevasses, limited to a porous firn layer, that have migrated inland through time. The communication of the results is somewhat scattered, with some statements in the discussion and conclusions not fitting with the interpretation of your results. Figures need to be made clearer, data availability and repository need to be checked. Three major comments:**

1. **Provide some additional context/clarity on the methodology for identifying crevasses on grounded ice. The lack of basal stress present in ice shelves allows for an ideal and accurate assessment of the damaged state of the ice; this is the reason why previous work by Lai et al., 2020, Izeboud et al. 2023, Surawy-Stepney et al., 2023, focused their attention on ice shelves rather than grounded ice (no presence of grounded ice friction). Can you comment on that and add some clarification in the paper on why you focus on grounded ice?**

*We focused on grounded ice in part because (as you note) several studies have already identified crevasses across ice shelves. More importantly, we also chose to focus on grounded ice regions as this is where the largest expansions in crevassed area and increase in crevasse density occur. Changes in the crevasse density over ice shelves are smaller in the Amundsen Sea Embayment (ASE) and include full thickness rifts (which we discuss in section 4.4.2 and have been discussed by Alley et al. (2021) and Benn et al. (2022), among others) in addition to the surface crevasse features that we highlight in this paper. We've added another sentence to the first paragraph that better recognizes past work on Neural Network crevasse detection (Izeboud et al., 2023; Surwey et la., 2023a, 2023b), and the relevance of grounded ice crevasse area change as it relates to theories of grounded ice fracture-based instability.*

2. **What are the problems with terrain shadows when assessing crevasses on grounded ice?**

*We are also careful throughout the study to avoid claims that we identify all crevasses on the grounded portions of Amundsen Sea glaciers. In fact, we demonstrate*

*that SAR imagery is not able to image some crevasses that we do observe in ice-penetrating radar. We identify two primary reasons that ice-penetrating radar: (1) the crevasse may be buried by subsequent accumulation as they advect downflow until the radar wave transmitted by satellite no longer penetrates to the now buried, and likely snow/firn filled, crevasse, and (2) angle of incidence of the satellite radar wave relative to the crevasse may result in unfavorable geometries so the crevasse sidewalls. High incidence angles may result in poor reflections off sidewalls. Crevasses with orientations parallel to the look angle of the satellite may also result in poor reflections off crevasse sidewalls. Neither of these effects is directly analogous to terrain shadowing as usually interpreted in satellite imagery collected in regions of steep topography, which only occur near a few large mountains (nunataks) in the ASE. However, these effects are somewhat similar to terrain shadowing in that they are geometric.*

3. **It would be good to understand how this methodology compares to the other available crevasse maps (Lai et al., 2020, Izeboud et al. 2023, Surawy-Stepney et al., 2023)? As for the map of crevasses, these tools should be made available online as they would be useful to other researchers in this field, are you planning to do that? (the github repository is empty). In your conclusions, you state: "Finally, the ongoing collection of SAR images can also be processed rapidly enough using our automated framework that live crevasse detection in areas where researchers are conducting fieldwork is possible."; how is this automated framework better than already available methods (Izeboud et al., 2023, Surawy-Stepney et al., 2023)?**

*We see these studies as complementary – by comparing algorithmic approaches with two deep learning methods that use different training sets, we can start to evaluate the relative skill of these models and converge on an approach that maps crevasses with the highest fidelity. We do not suggest that this automated framework is better than other implementations of SAR imagery segmentation, which in most cases are using a very similar approach. Multiple groups were working on similar approaches as we were finishing our analysis and writing this paper. We change the final sentence of the paper to acknowledge available methods*

*"Finally, the ongoing collection of SAR images can also be processed rapidly enough using automated framework (e.g., Izeboud et al., 2023; Surawy-stepney et al., 2023a; Surawy-stepney et al., 2023b; and our work) that live crevasse detection in areas where researchers are conducting fieldwork is possible"*

*We also hope our approach, which evaluates identified crevasses against other companion data (like ice penetrating radar and high-resolution optical imagery) and attempts to identify the impact of surface crevasses on ice mechanics while also providing quantitative links to earlier studies (Vaughan et al., 1993) provides a useful new framing for the community.*

*The methods used will be published in the github repository and archived on zenodo at the time of publication. We are not field safety experts and are not equipped to produce maps of hazard for logistic purposes, but we will initiate conversations with field safety personnel on how best to use these results.*

4. **Implications of firn crevasses: At the end of your discussion, you say, "Surface crevasses also affect estimates of ice flux across the grounding zone." However, previously, you mentioned that your results suggest that surface crevasses expressed in satellite remote sensing datasets only weakly affect the bulk viscosity of ice shelves, and many times, throughout your text, you say that these features do not affect flow. It is important to remember that only those surface features that actively influence ice flow are pertinent to ice-flow dynamics and changes in grounding line flux. In the conclusion, you say: "The time series of crevassed-area evolution that we use to support these results presents a valuable target for models that incorporate near-surface fracture or continuum damage mechanics." While I think they are useful maps, they do not necessarily help modellers when assessing damage, as they are such shallow features that do not affect the bulk ice viscosity. Moreover, If they were to propagate, ice shelves still bear enough buttressing capacity (Gerli et al., 2023a). I would discuss more about the local implications of these surface firn crevasse features (you already talk about the uncertainty in mass balance which is great to see). You could add something in terms of the local implications: increase of surface ice roughness, which enhances solar radiation and reflection in the surroundings and promotes atmospheric turbulent heat fluxes, all of which intensify melting at the ice surface, causing firn saturation, meltwater ponding and potential risk of hydrofracturing.**

*We realize now our description and its intent could be clearer. We were not intending to suggest that surface crevasses affect estimates of ice flux across the grounding zone through their role in changing ice dynamics, but rather, crevasses are void space that are not factored into observational studies of surface mass balance, such as Rignot et al. (2019) and Sutterley et al. (2014). We have added citations in the text to clarify this connection. If there is void space that isn't accounted for in the near surface associated*

*with crevasse void space, then absolute ice-discharge fluxes will be overestimated. If the spacing of the crevasses or their penetration depth also change (leading to a change in void space) then changes in the discharge flux could also be misattributed. The fraction of the void space is small, but it may increase independently of the discharge with interior changes in effective stress and could be included as a focus in large-scale monitoring programs that seek to better resolve absolute ice discharge, which currently focuses on improving the resolution of bed topography at the periphery of the continent to accurately estimate absolute discharge (for example, the Rings project [https://scar.org/science/cross/rings](https://scar.org/science/cross/rings)).*

*Regarding the statement that only those surface features that actively influence ice flow are pertinent to ice-flow dynamics and changes in groundling line flux, we agree with the reviewer. Hopefully, the above paragraph makes the distinction between observational studies that quantify discharge flux and then independent constraints these observations may (or may not) provide for damage models. We now turn to this second point the review makes. We agree with the reviewer that most crevasses on grounded ice likely have a limited impact on the effective viscosity of the ice because they are near surface features. We have deleted the sentence in the conclusion that referenced implications for ice-flow models.*

*The reviewer also suggested that we discuss literature on the impact of crevasses on surface albedo. The point is really interesting, and we think it deserves more complete treatment and consideration for zenith angles that are appropriate for Antarctica. The work of both Pfieffer and Bretherton (1984) and the Cathles IV et al. (2011) have been cited in a new section added to the end of the discussion titled Surface Albedo:*

*"The appearance of surface crevasse features in panchromatic and SAR imagery also highlights the effect of crevasses on ice-sheet albedo. Incipient roughness associated with near surface crevasses has previously been suggested as potential mechanism for radiatively driven surface ablation feedbacks that can cause small-scale roughness features to grow and eventually impact the crevasse morphology and surface meltwater storage (Pfeffer et al., 1987, MacClune et al., 2003, Cathles IV et al., 2013). The growth and evolution of surface features in mountain glacier environments, such as sun cups and penitents (pointed ice or snow columns) has also been connected to solar absorption feedbacks with surface roughness (Betterton et al., 2001, Warren et al., 2022). Previous studies have used radiative transfer models to understand how the shape of these features affects their evolution. These studies have shown that crevasses of reasonable geometries for latitudes consistent with Greenland outlet glaciers can increase the local absorption by to >50% (Warren et al. 2022). Fewer studies have explored this feedback for crevasse orientations and latitudes consistent*

*with Antarctic outlet glaciers for present and warmer future climates. We leave this as a promising extension of the work presented here."*

5. **Perhaps future work? It's a shame you don't investigate more the presence of basal crevasses in the ice shelf.**

*There is a great deal of literature on basal crevasses and a growing archive of ice-penetrating radar data that should be used to understand how these features change due to its ability to image basal crevasses that do not penetrate to the surface. The penetration depth of these features particularly on the Thwaites ice shelf is remarkable, and we are excited to continue to think about fracture using radar datasets and satellite imagery in this region in separate, future work.*

**Specific comments**

**Line 131 – You use the F1 score to choose a threshold for the binary classification (greater than 0.8) ? Is a typical thing to do? Can you add references? I see the f1-score vs threshold plot in Figure S2; it would be good if you could add more information in the caption.**

*We have added references for the use of the F1 score. We've also added a section in the supplement explaining what the F1 score does.*

*"The U-net we implemented includes 3 down-sampling layers, and 3 up-sampling layers that were trained on 1600 images from the Amundsen Sea Embayment. The trained network takes preprocessed input images and produces images with pixel-wise probability for each individual pixel to include a crevasse. These probabilities are used to create binary detection masks using the F1-score on independent training data. The F1-score is a measure of the accuracy of binary classification algorithms and was used to diagnose the precision and recall of pixel-wise crevasse classification based on a threshold probability. The F1-score can be written formally as:*

*F1 = TP/(TP+½\*(FP+FN))*

*where TP is the number of true positives, FP is the number of false positives, and FN is the number of false negatives."*

**Line 237 – misspell Interpreted**

*Thank you for this catch. We appreciate the gentle revision.*

**Line 264 : Section 2.3?**

*This is part of the results section (see line 230) and thus is 3.2.*

**Line 394-5: "These crevasses would not be expected to penetrate deeper into the ice column." Expand and add references.**

*We have changed this sentence to:*

*"These crevasses would not be expected to penetrate deeper into the ice column where the denser ice has a higher tensile strength  (Rist et al., 2002) and where the increased pressure with depth (glaciostatic pressure) is predicted to close dry air crevasses (van der Veen, 1998)."*

*For both grounded glaciers and floating ice shelves, crevasse depths predicted by linear elastic fracture mechanics do not penetrate the full thickness of the ice column. That they open at all is a reflection of the tensile strength of firn rather than ice and that is the point that we make at the end of this paragraph.*

**Line 458-9: "Our results indicate that changes in the crevassed area are generally correlated with increases in surface stresses in response to ongoing ocean-driven acceleration." Why do you say that? Are you showing any correlation to ongoing ocean-driven acceleration in your results?**

*The effective strain rates on Thwaites are increasing as the glacier thins and accelerates in response to increased ocean melt. We assume that the prefactor in Glen's flow law and the nonlinear exponent in this law are fixed (which is an assumption that we have more clearly described in the methods sections). The principal stresses thus also increase due to increase in longitudinal strain rates. These ellipses show a consistent pattern over time. When effective stresses exceed 82kPa fractures occur, below this effective stress, the ice sheet remains mostly uncrevassed. Therefore, we conclude that the crevassed area is increasing in response to increases in acceleration. We have removed the ocean driven component as this link comes primarily through literature outside of this study.*

**Line 465: "At present, the near-surface crevasse features appear to be a symptom rather than a driver of acceleration and retreat in the ASE." Explain why is the case.**

*The features that we observe are consistent with crevasses that only penetrate through the near surface of the firn/ice column. Because our modeling and observations suggest that these are near surface features and as you point out earlier in the review, there is a limited effect they can have on the bulk fluidity of the ice column. Previous studies have used fracture criterion for ice based on observations of surface fractures that likely*

*reflect the material strength of polar firn and not ice. These models have been used to suggest that material failure in areas like the ASE can lead to runaway retreat and have led to a lot of speculation that surface fracture is connected to the stability of these glaciers; in actuality, this relationship is weak (Greili et al. 2023). We have changed this sentence to say:*

*"We speculate that at present, because of the near surface nature of these features, surface crevassed area change appears to be a symptom rather than a driver acceleration and retreat in the ASE. Because these are near-surface features, there is a limited effect they can have on the bulk fluidity of the ice column"*

**Figure comments**

**Figure 3 : Would it be possible to outline the major crevasses or make a box-area for crevasses that are visible in SAR, and not visible in the panchromatic imagery? It would help the reader when you are comparing the two products.**

*We have outlined Figure 3 using the expression of crevasses in Worldview and high-resolution (0.5 m) optical imagery and Sentinel satellite SAR imagery. We have included two versions for the reviewer to consider here one with the outline as the reviewer suggested and then another with a heat map of crevasse probability produced by the classification routine. We prefer version 2, which we have included in the text.*

[Figure]

*Figure 3 v1*

[Figure]

*Figure 3 v2.*

**Figure 4, enhance color visibility of panels b and c, red crevasses are barely visible. Panel d and e could have a higher contrast (especially panel d) for the detection of buried crevasses.**

[Figure]

*We have tried to balance making the identified crevasses transparent so that the underlying features that aren't visible in the WV image are more apparent.The higher contrast saturates the surface. We prefer using the color scale that is shown. These*

*radar images are not calibrated to absolute power and thus the dB scale varies between images. Even if it was set to a standard scale, it wouldn't be referenced, so the values would be meaningless. Referencing to an absolute scale could be helpful for an objective subsurface crevasse detection scheme, but this is beyond the scope of what we are doing in this study.*

**Figure 5 : Misspelling "in"**

*Thank you for this catch.*

**Figure 6: use a different colour scheme with better contrast; it is really hard to see the mapped crevasses**

*We will use a different color scheme. Thank you.*

**Figure 7 This is an interesting figure. I would be interested in seeing the location of these crevasses where stresses are greater than 200Kpa. Perhaps you can choose a year and represent their spatial distribution and colour crevasses as a function of stress?**

*We have added Figure 9, which shows the spatial distribution of the von Mises stress criterion for quarter 1 (January to March) 2019 and the location of crevasses as well as the histograms of the von Mises stress criterion for pixels classified as crevassed and uncrevassed.*

**Figure 8 red lines in c) are the same as the dotted lines in a and b?**

*Yes, this is correct. We have made this more clear in the text. See additional parenthetical below:*

*(dotted lines in panels a and b).*

**Figure 10 subpanel c and d) I would colour the x-axis by the colour of the transect in subpanels a) and b) to make it easier to visualize. You can point out the major crevasse event in a) and b) that you talk about in the caption.**

We have colored the frames of the sub panels for the radargrams to correspond to the colors of lines in the map panels. The major crevasse event is shown in a zoomed in window below.

[Figure]

**Figure S3 and S4 could be improved by adding some coloured lines to map features that correspond to both SAR and Optimal Imagery.**

*We have added colored boxes to map features that correspond to both SAR and optical imagery. We've also included boxes that effectively show the full resolution (zoomed in) of the Worldview imagery.*

[Figure]

**Figure S5 subpanels b) and d) have difficult readability; crevasse mapping needs better contrast.**

*We have changed the mapping contrast to show areas where probability is high in black.*

[Figure]

**Figure S6 a) and b) you could add the grounding line position for reference. c) and d) where are the black arrows?**

*We've added grounding-line positions that correspond to the times of imagery acquisitions in this study using the dates of the reported grounding line in MEaSUREs (Rignot et al., 2016, updated 2024). The black arrows are intended to highlight the*

*maximum height of basal crevasses.*

[Figure]

**I really like Figure S7, especially the radargrams of subpanels c) and d). Again just for reference I would add a grounding line position for subpanels a) and b) .**

*Thanks! We have added a grounding line to the figure.*

[Figure]

**Line 346, 365 and 369-370 you mention figure S8 and S9, but I imagine you meant figure S7? Or are figures S8 and S9 missing?**

*Yes, correct, thank you for catching this mistake.*

**General comments from reviewer 2:**

In general, some more work is required to demonstrate that the result in the title is widespread. This could be improved by showing a greater number of crevasse maps, rather than focusing on a small portion of Thwaites Glacier (as in Figure 6) and some large scale statistics like crevassed surface area (as in figure 7b-c).

The inland propagation of crevasses is shown in the main paper only in figure 6, where we can see some evolution on the south-eastern side of the crevasse field between March and June 2019. However, the larger region in Figure 1b (where the colours are labelled the other way round, or perhaps incorrectly?) shows a more varied picture with some locations showing that crevasses have been advected downstream and not replaced, and others where crevasses have developed further upstream. Figure 7c shows that the crevassed area on Pine Island Glacier (imaged by path:53, frame:857) increased by 25% between 2016 and 2023. This is a big change so it would be great to see some maps showing the difference in crevasse area at the start and end of the timeseries, especially as there is no indication of uncertainty in the plots.

There is quite a lot of talk about correlation of increased crevasse area with changes in speed or increased surface strain rates. However, we are not presented with any data regarding changes in speed or surface strain rates in the ASE and no analysis linking this I found section 2.3 a bit difficult to read, and the diagram in Fig. 2 is not particularly illuminating. However, the idea is pretty simple and I think can be reduced to something like the following argument:

*For Figure 7, we have made the markers for each signal different (previously these markers were the same with close enough colors that two were indistinguishable). These markers allow the reader to better see that there aren't dramatic changes in the PIG record at the end of 2016. We've also used 82kPa as the threshold for the von Mises stress criterion for area change instead of 200kPa. We've also included the change in the density of crevasses during the period of Pine Island crevassed area at the beginning of the PIG record in the supplement (S8). This figure shows the crevasse area change and the von Mises stress calculated from velocity products from 2015 and 2023 showing an increase in both the area of crevasses and the von Mises stress over*

*the observational record.*

[Figure]

The porosity P of the firn is defined as the ratio of the volume of air contained in the pores to the total volume. The fractional area of air intersected by a plane is then ~ P 2/3. Hence, the tensile strength of the firn, which is proportional to the area of ice intersected by a plane normal to the background tensile stress, is reduced by a factor ~ (1 − P 2/3) compared to firn of zero porosity. At high porosities, some of the filaments of ice separating pores from each other break and the pores can merge. To account for this, we follow Jelitto and Schnewider (2018), and introduce an additional factor (1 − P )n where n is a parameter we fit to data from laboratory experiments. Hence, we find:

σf/σl= (1 − P )n × (1 − P 2/3).

**This model does not constitute a major novelty of the article (as it is described in Jelitto and Schnewider (2018) and is not validated by tensile strength estimates later) so this argument could be moved into the discussion where it could sit alongside figure 9.**

**However, I am not sure the description of the firn model and figure 9 add much to the article, so would just suggest removing them. The porosity of the firn is not known (at least, it is not shown in the article) so what figure 9 serves to do is show that the tensile strengths found from the estimates of surface stress are not inconsistent with firn described by such a model. This is fine except that these data are compatible with pretty much any model where tensile strength decreases monotonically with porosity (especially models with two fitting parameters). The fact that we do not know the porosity of the firn means that the strengths calculated from the satellite data do not provide evidence for the model.**

**Furthermore, the model itself does not provide any further evidence for the crevasses initiating in firn than figure 8c plus the statement that firn tensile strength should decrease with porosity. Overall, in my opinion, there is not really a need for this in the article so I would recommend taking these sections out. I think figure 8c is useful and interesting, and does the job at demonstrating that the crevasses are likely to initiate in firn. This might have the added benefit of improving the overall flow of the article.**

*Although we agree that the data presented here cannot directly prove the model we suggest, we think this section is important to include here because there still exists a disconnect between porous fracture studies in material science and the communities that study firn where observations have primarily been focused on regions where strain rates are low and fracture (crevassing) does not occur. As a result, we have elected to keep this in the article, but we highlight the important point that you make: that this model is just one among many which could all be consistent with our data. We also more clearly state how the work extends from analytic descriptions of tensile strength from porous materials research by Jelito et al. (2018). Even though we cannot show the presented model is uniquely suited to reproduce the data, we think it is important that we include a firn air model and shows that these models of firn air content across the basin agree with the porosities that we describe in our model.*

*For figure 7, we've also included the change in the density of crevasses during the period of pine island crevasse growth at the beginning of the PIG record in the supplement. We also note that the PIG timeseries of crevasse area change from 2016-2018 figure 7b includes two different frames (frame 856 and frame 857) that have colors that were previously difficult to distinguish. We've added a supplemental figure, Fig. S9, that also shows this change in crevassed areas.*

**The ellipses shown in figures 7 and 8 look great. It is evident that the vast majority of the crevassed locations on Thwaites sit inside the bounds of the yield region shown, and the vast majority of the un-crevassed locations sit at effective stresses lower than the inner yield curve. There are a couple of things I hope the authors can elaborate on.**

**Firstly, the ellipses show that reaching a stress threshold between 70 and 200 kPa is a necessary condition for fracturing. However, I wonder whether the authors could comment on whether it is a sufficient condition? The panels 8a and 8b have different scales on the colourbars (necessary because of the huge class imbalance between crevassed and un-crevassed locations), so it is not clear whether a measured surface stress between 70 and 200 kPa is a good indicator of whether there will be a crevasse. Perhaps the authors could show a map of where**

stresses in the Amundsen Sea Sector exceed 200 kPa. The figures at the moment do not rule out the possibility that the von Mises stress criterion is a good way of determining where crevasses are not likely to be, but might not be particularly skillful at determining where crevasses are likely to be (potentially because the strength of the ice varies too much due to differences in firn porosity?). If this turns out to be true, it is an interesting point of discussion.

*This point was made by another reviewer. We believe that the reviewer's conclusion is correct: the upper bound that defines the elliptical structure of the crevasses may reveal more about the distribution of stress on Thwaites at present rather than anything about the fracture limits of ice. The von Mises criterion indeed does not constrain or limit the critical stress at which ice fails.*

It is also not clear exactly which area of Thwaites Glacier is being used to test the stress envelopes. Is it the entire crevassed area on Thwaites or the small section at the upstream edge of the crevasse field? It would be great to see this analysis extended to the rest of the Amundsen Sea Sector (we are told other glaciers show similar results). If the concern is that the porosity over the area has to be relatively similar to get a well-defined stress envelope, then the area could be broken up into smaller sub-regions. Given the potential here it seems a shame to limit the conclusions to a particular glacier rather than saying something more general about the failure behaviour of ice.

*We have added Figure 9 to shown which frame was used to construct the failure envelopes and the area of the frame (white dashed domain in Figure 9a) where the crevasse classification routine is applied. There are also two new figures in the supplement to show what these ellipses look like for pine island and the glaciers that feed the Dotson and Crosson ice shelf. They are much the same, and consistent with similar ellipses determined for Greenland and Antarctic outlet glaciers (Vaughen et al. 1993).*

1. I would use the word "ellispe" or "elliptic" throughout the article when describing the von Mises stress envelopes, rather than "ovoid/ovaloid" unless that's standard.

*We have changed ovaloid and ovid to ellipse and elliptic throughout. Thank you for the suggestion.*

2. Figure 1. I think the colours for panels b and c might be labelled the wrong way round. If not, it would be better if they were consistent with figure 6.

*These panels are labeled incorrectly thanks for the catch. The panels are now labeled correctly.*

**3. Line 90: It is not clear to me why the use of Sentinel-1B images would result in geometric inconsistency.**

*The sentinel-1B images were not used because the record of 1B imagery is not as long and because the geometric domains of the scenes were different. We were careful when using images to compare frames that were consistent in look angle and geometry across the timeseries to minimize false positive identification of crevasses that are sensitive to the orbit geometry relative to the sloped surface.*

**4. Line 104: Is there any scaling applied when the images are converted to 8-bit integers, for example by something like requiring the mean backscatter to correspond to a value of 100. Otherwise, do the values concentrate at 0?**

*There isn't any scaling here, and we do have some concentration of values near 0 in histograms of the amplitude for the entire image.*

**5. Line 127: I would write "Adam" optimisation rather than "Adams" optimisation.**

*We have changed the wording to Adam. Thank you.*

**6. Line 137: This part about how the images were combined - including the statement about the translational equivariance - is a little unclear to me. Is it the case that the padding is applied to an image patch prior to its processing by the neural network, and that the processed image could be in a different location?**

*The images are combined after the identification. Padding is used to limit edge effects associated with features at the boundaries of the original image. These methods are common in the image segmentation literature where the images are much larger than the images that U-net can readily use for training (typically 500x500 pixels).*

**7. Line 237: "interpretted" should be "interpreted**

*We appreciate the gentle correction.*

**8. Line 244: Figure 4b seems to be referenced both as evidence for the visibility and invisibility of crevasse features in optical imagery. Perhaps this should be 4c?**

*In this figure, we show how crevasses appear in surface imagery. This is the imagery regularly used by SAR teams to evaluate large-scale surface hazards in imagery before*

*traverse teams are sent in and the literature/ use of these features in science is still primarily qualitative. Here, Figure 4b is intended to demonstrate that crevasse may be present but not easily visible in optical imagery. Figure 4c is intended to demonstrate that the crevasse classification would be consistent with inspection of optical imagery showing no crevasses in this location.*

**9. Line 276: Could you provide a citation for the 500 ma−1 increase in speed deep into the interior of the Pine Island Glacier between 2017 and 2022?**

*Yes, this has been discussed by Joughin et al. (2021). See figure from this study below.*

[Figure]

**Fig. 1. PIG location map and changes in flow speed over the past decade.** (**A**) Locations of points where speed is sampled (GL−2 and GL+2), moorings were deployed (PIG N and PIG S), and centerline profile (gray) over a 2019 velocity map of PIG. Black box indicates area shown in Fig. 2. (**B**) Time series of speed at points ~2 km upstream (GL−2) and downstream (GL+2) of the grounding line derived from SAR data collected by the TerraSAR-X/TanDEM-X (TSX) and Copernicus Sentinel 1A/B (S1) missions. The 90-day moving average of mean 450 m-650 m depth ocean temperatures from moorings located toward the north (PIG N) and south (PIG S) ends of the shelf front are shown (*17, 18*). (**C**) Speeds along centerline profile. Dashed, color-coded lines indicate locations of the GL±2 points.

**10. Figure 6 is very nice, but it is not really discussed in the main text. Perhaps some details specific to the figure should be discussed in section 3.2.**

*We appreciate this suggestion and have now taken more time to cite Figure 6 in section 3.2. Specifically, in the bulleted list in lines R278-285 that summarize how crevasses evolve in the imagery time series presented in Figure 6.*

**11. I think Figures 6-8 could be restructured to make them more compelling. For example, it would be great to see a before and after image for Pine Island, Thwaites and Pope/Smith/Kholer along the lines of figure 6 to accompany the timeseries of crevasse area change in Figure 7. It is a bit of a pain to go back to figure 1 to see the path/frame numbers when looking at figure 7b/c, so would it be possible to reproduce the acquisition footprints here as well? I think it would be better to either make 7a its own figure or combine it with figure 8 to improve the focus of this part of the article. As it stands, these sections seem to jump about a bit and lack focus to some extent.**

*We agree with the reviewer here and have added a figure in the supplement similar to figure one at the beginning and end of the timeseries to show the progressive change in crevassed area on Pine Island Glacier.*

**12. Figure 7a: Could the authors comment on why the density of pixels seems to change from year to year at the same location? For example, the densities seems uniformly lower on Thwaites in 2022 than 2021. The corresponding crevasse area time series in 7b show that the total number of datapoints should be roughly the same so the datapoints are more widely distributed? What causes this?**

*This is the result of sampling of the velocities, which are not seamless in time varying due to the quality of the data that was used to construct the velocity fields (both Sentinel-1A and Sentinel-1B data to start and then just Sentinel-1A and Sentinel-1B failed).*

**13. Figure 7a: There are some faint dashed lines in the figure which, in the top row at least, indicate the coordinate axes σ1,2. These seem to change position in years 2017-2020. Given the statement on line 290 I guess this is a mistake?**

*Apologies for the resizing errors. These plots have been made with the same axes.*

**14. Line 292: Is this calculated as an average over the whole timeseries of crevasse maps and velocity fields or does it correspond to a specific date? If the latter, why was it chosen?**

*Velocity data for this region were available at quarterly resolution and define the interval that we use for the crevasse data.*

15. Line 294 and Figure 8: Can the authors elaborate on how the dashed curves were

calculated as these form the basis for the observed strengths. For example, are they

the closest ellipses to a contour on to the density plots (10% of the maximum density

defined over some area could be an option)? There is also some inconsistency between how the bounds are reported in the article - sometimes as 75 − 210 kPa and other times as 70 − 200 kPa.

*These were subjective bounds. We have now done as the reviewer suggested. 90% of crevasses occur above the lower bound ellipse, which corresponds to 82 kPa. 90% of the crevassed pixels occur below the upper bound ellipse, which corresponds to 165 kPa. These values were calculated including all velocity scenes and crevasse classifications (all quarterly images from 2015-2022, inclusive). We have also added a sentence describing the meaning of the lower bound for effective stress (R310-R312).*

**16. Figure 9: Could you include in the caption that the red lines are the same as those in the previous figure?**

*Yes. This has been added to Figure 9. Thank you for the suggestion!*

**17. Section 4.4.2: This section is not quite right. The maps of fractures in the studies referenced deal in large part with the surface expressions of ice shelf rifts and basal crevasses, not with the kinds of crevasses considered in this article. It is true that the surface crevasses seen here, when advected onto the ice shelves do not change a great deal but this has little relevance to the other studies referenced here, or to the work of Gerli et al., 2023. I would recommend removing this section and adding a sentence to the end of the first paragraph at the start of section 4.4 explaining that your results show crevasses of this type have no relevance to ice shelf weakening.**

*The maps of fractures in the studies referenced here deal with rifts and the expression of basal crevasses in surface imagery (we note this doesn't necessarily include all basal crevasses). They also explicitly include the surface crevasses that are the focus of this study. Surface crevasses visualized here, when advected onto the ice shelves do not change the fluidity. The radar data that we present as part of this study show that the features that appear in surface elevation data show that the depth of the basal crevasses varies.*

**18. Line 480: "surface strain" should be "surface strain rate".**

*Thank you for the suggestion. We have made the suggested revision.*

**I very much enjoyed this article delineating crevasses in the Amundsen Sector. The delineation approach is quite novel, and the time-space analysis of crevasse changes is also quite novel. I share some thoughts on a few general points below. Beyond**

methodology and results, I also enjoyed the interpretation, although I think more support is needed before stating so conclusively that the crevasses are limited to the firn and have no dynamic impact on ice flow.

**Firn-only -- The crevasses are referred to be "restricted to the firn" several times, including at abstract level. I would like to see further context for this conclusion. For example, what is the pore close-off depth in the region (i.e. firn-ice transition)? Simply put, how deep is the firn? Related to this, Figure 4e shows very deep crevasse tops (i.e. 130 m deep?). It is said that these may be off-nadir crevasses that are actually shallower, but in terms of a first-order calculation, how off-nadir would these crevasses need to be for the geometry to be projected to that depth? Is that reasonable? At present, the reader is uncertain how deep the firn extends, and whether crevasse tops are deeper than this.**

*We appreciate the reviewer's criticisms of the interpretation of these features as near surface. We have included more context using the density data from Gow et al. 2004. This figure is in the supplement. And shows that the firn-ice transition in the upstream regions is at most 120m depth. This is an upper bound on the depth of the firn-ice transition downstream that has experienced significant horizontal thinning (Horlings et al. 2020) compared to the upstream sites of these firn cores.*

*The features we link to crevasses have been SAR focused. This processing migrates energy that is collected in the along-track direction. In unmigrated data these features appear as hyperbolic reflections connoting the $r^2$ dependence of the scatterer in off-nadir traces. In some regions the crevasses do extend below this depth and suggest that small inclusions persist below the firn ice transition that may be inherited from near surface fractures. These deep crevasse features (below the firn-ice transition) have been in the Ronne and the Ross where they have been interpreted as buried crevasses (i.e., Kingslake et al. 2018).*

[Figure]

*See figure below from data collected near Ridge A, for instance.*

[Figure]

**C-band penetration -- As written, the paper seems to overestimate the penetration depth of C-band SAR, saying "C-band SAR imagery penetrates up to several dozen meters into the subsurface." My understanding of the cited Rigot2001 is max 10 m penetration depth. Jezek only says "several meters" https://doi.org/10.3189/172756499781820969 . "Several dozen meters" seems to be quite on the high side, although I am not sure what impact, if any, the depth of penetration would have on analysis.**

*We have edited this to reflect this change. We agree that several dozen meters penetration may only occur in some special conditions and is too high a number to be stated here. We have reworded to several meters. We do note that several of the features we observe in radar data appear more than 20m below the surface (sometimes closer to 40-50m). We have made the suggested change but wonder whether the reviewer has thoughts on how we image these deeper features that run orthogonal to the radar profile and therefore are imaged at nadir.*

**Crevasse width – I suppose it is implicit that the algorithm only detects crevasses wider than ~10m, or the Sentinel pixel size, or is it the resampled ~25m pixel size?**

**These are clearly large crevasses. Is it possible that there are smaller, or narrower, crevasses that go undetected by the algorithm? Or put another way, can the authors say something about lower limit of crevasse geometry down to which they have detected? Presumably these maps would be a lower limit on the damage extent of ASE, if some scale of smaller crevasses and fractures has gone unmapped.**

*The reviewer is correct that we use imagery that has been resampled to a standard 25 m pixel size. However, the pixel size does not represent a limit for crevasse detection size. Since the radar imagery is sensitive to reflections off crevasse sidewalls, a pixel may be very bright if it images a reflection from a sidewall in a favorable geometry even though the crevasse itself is much smaller than the area represented by the pixel. Narrow-width crevasses with sidewalls orthogonal to the incident radar wave may be bright reflections even if the crevasse width is small. Using the data presented here, we cannot clearly relate the crevasses in SAR imagery directly to their true geometric size, so we prefer not to speculate on the true size of the imaged crevasse. We simply refer to detectability and changes in crevasse locations in the regions where we can detect crevasses. We think this point is important and now discuss it more extensively in lines 245-249.*

*"Optical imagery and ground penetrating imagery also reveal that the pixel resolution of crevassing in SAR imagery resampled to $25m^2$ is not directly related to the size of the detected crevasses. Since the radar imagery is sensitive to reflections off crevasse sidewalls, a pixel may be very bright if it images a reflection from a sidewall in a favorable geometry even though the crevasse itself is much smaller than the area represented by the pixel Marsh et al., 2021. Narrow-width crevasses with sidewalls orthogonal to the incident radar wave may appear as bright reflections even if the crevasse width is small. Using the data presented here, we cannot clearly relate the crevasses in SAR imagery directly to their true geometric size, and therefore restrict interpretations of crevasse appearance to changes in crevassed area in the same Sentinel-1A image scene."*

**Firn Fracture Model – I like the idea of modelling firn fracture, as this is not often done. I guess there should be an explicit assumption stated that firn properties are constant over the 2017-2022 epoch of crevasse migration. Presumably, if the firn properties are changing through time (i.e. firn becoming increasingly brittle due to refrozen ice layers) then this can also impact apparent crevasse extent, without changes in the underlying ice stresses. This paper implicitly assumes that only dynamic stresses have changed, not firn properties. This is fair, but should be made explicit and perhaps discussed.**

*We have made this assumption clearer. The assumption that firn properties have changed is something that should be evaluated with compaction velocity measurements from ApRES, but these data have yet to be collected in the Amundsen Sea Embayment. The*

*model we use suggests that refrozen firn strengthens the near surface (i.e., the firn would become stronger under tension with more ice layers). These variations with depth have a very limited impact on the bulk strength of the material as they scale with the thickness of the ice lens.*

**Limited Direct Impact – I have some difficulty accepted that there is limited dynamic impact from these crevasses. The ice thicknesses only look ~300 m thick in Figure 4, and the authors have only delineated the top of the crevasses. Even if the crevasses only extend 50 m deep, that is still 1/6 ice thickness. By the provided van der Veen citation, it is conceivable they could be 100 m (1/3 thickness) deep. Presumably some ice dynamic model parameterized with and without such crevasse geometry/prevalence is needed to state so conclusively that such large crevasses are not impacting relatively thin ice?**

*We apologize for presenting a potentially confusing radargram here. The ice thickness in the profile is ~2 km and we do not show the full thickness radargram that includes the basal reflection in this figure. This makes the crevasse reflections we intend to emphasize here difficult to see. The 300-m depth return is the surface multiple. We have added a statement explaining this to the figure caption so that it is clear to readers. Because the surface is porous and the pressure condition here is zero, the maximum difference in overburden is ~10 kPa, which is small compared to the uncertainty for instance in friction proxies at the ice bed interface.*

**Failure Envelopes – Figure 7a is too small to be useful. I'm not sure if presenting 32 failure envelopes is the best thing for the reader. Perhaps anomalies, by either the eight years and/or four seasons, might be more informative to highlight differences and change.**

*We appreciate that there is a lot of information in that figure, and it can be difficult to parse at first glance. Our goal here was for this figure to be used for detailed reference rather than high level summary. While it is a slightly different purpose than the figure you are suggesting, we prefer to keep this figure in for any readers who want to see a comprehensive review of the inferred stresses.*

**In Figure 8, are you calculating the "crevassed" and "uncrevassed" pixels at the 25 m resample pixel size? Presumably there would be "uncrevassed" pixels between individual crevasses. So, are you averaging over some distance? Perhaps visualizing a binary crevassed/uncrevassed map would be a helpful inset here.**

*This is a great point that was also brought up by one of the other reviewers. Because we resample the images using a nearest-neighbor algorithm to preserve abruptness, it is possible that crevasses that are smaller than the 25 m pixel size (but larger than the*

*minimum 5 m pixel side length) are preserved and detected by our approach. Narrow-width crevasses with sidewalls orthogonal to the incident radar wave may appear as bright reflections even if the crevasse width is small. But to your point, the most detectable features by this approach will be > 25 m, and smaller features would be missed. We've added a paragraph discussed above to the results section that highlights our considerations when we interpret crevasse width.*

**Tensile Strength – Figure 9: I guess there is more data than this available. See, for example, https://erdc-library.erdc.dren.mil/jspui/handle/11681/2698, which comes to mind. At the moment, misfit between remotely sensed crevasses and measured tensile strengths is attributed to "porosity dependence of near-surface tensile strength". I wonder if there may also be some effects of anisotropic fabric, even in the near surface firn?**

*Anisotropic damage evolution relations have been developed and compared with observations of rift propagation using along flow (Jimenez et al., 2021) and depth average flow models (Huth et al., 2023). From an ice core and firn cores collected in Northeastern Greenland, fabric anisotropy does develop quickly in areas where horizontal strain rates are high. There is more work that could be done in the laboratory to understand how tensile strength depends on the crystal fabric orientation relative to the applied stress, but that is beyond the scope of this paper as we do not present data that can address anisotropy.*

**Fracture Mechanism – Anisotropic firn fabric would be most relevant for mixed-mode fracture. Perhaps firn reacts similarly to flow-aligned mode 1 opening across the study region, but if there is mixed-mode fracture, then crevasses aren't necessarily always associated with mode 1 opening aligned with fabric. It would be helpful to have some velocity-derived flowlines on a map with delineated crevasses, so the reader than see that flowlines generally intersect crevasses at 90° (i.e. characteristic of mode 1 opening).**

*Because we don't know the firn fabric, we assume the fabric in the firn is isotropic. Where firn cores and phase sensitive polarimetric radar measurements have measured fabric development in fast flowing environments (i.e. Zeising et al. 2022), most of the fabric anisotropy develops below 100 m. We can plot flowlines on these figures as the reviewer suggests. Because these features advect with the flow, their orientation does not necessarily preserve the orientation relative to the velocity field when they opened. The velocities of Amundsen Sea glaciers have also changed significantly over the last four decades. We can constrain some of this change, but satellite observations over this period become much sparser prior to 2015. Changes in the velocity field (particularly ongoing acceleration) may explain patterns we observe where crevasses appear to cross one another as new fractures open where preexisting fractures have been advected from*

*upstream (see for instance Fig. S9 panel a and b). We've also included an example of three crevasse features that we've backtraced according to the modern velocity field. These features appear to have opened parallel to ridges in basal shear stress upstream near the outlet of subglacial outflow from the largest subglacial lake on Thwaites Glacier.*

[Figure]

**Major remarks:**

1. **The geometric model is not new and not properly used in the manuscript. Its derivation and the design of Fig. 2b-c closely follow Jelitto and Schneider (2018), which is not adequately stated. The authors only have applied the geometric model for the tensile strength of porous materials from Jelitto and Schneider (2018) to polar firn but not developed it. In the conclusion, the authors write,** *"Using this new model, we showed that the increase in the crevassed area on Thwaites Glacier is consistent with the increase in the area where effective surface stresses exceeding a critical tensile strength for the near-surface firn."* **I do not see how the model is used in this way. The model is only tuned to laboratory data but not used any further. For example, why is it not applied to available firn data from Gow et al. (2004)? This could give a prediction of crevasse depth for a given stress field.**

*The model follows closely from Jelitto and Schneider (2019), but we solve for the tensile strength rather than the fracture toughness. We do recognize that the same substitution used in Jelitto et al. (2018) was used to determine the tensile strength and we have now cited this paper more thoroughly throughout this section, including where our treatment is different. Using the available firn data discussed by Gow et al. (2004), we show that the porosity dependence of the firn strength limits the penetration in these interior*

*regions where the ITASE surface traverse collected firn cores. We can add these figures to the text with a figure that shows the porosity dependent limits on penetration, but, for now, include them below.*

[Figure]

2. **The relation between increasing effective surface stresses and crevassed area is also elsewhere not clearly shown. Variations of flow speeds are only described in the text of Section 3.2. They could be displayed nicely along with data of the grounded crevasse area from Fig. 7. I assume the intention of Fig. 7c is to illustrate the flow dynamics, but the text never refers to it, and it is unclear why "grounded ice area where σe > 200 kPa" is chosen as the metric for illustrating this. Why is the threshold at 200 kPa and not 75 kPa, where crevassing initiates? I would suggest illustrating this relation by looking at the time series of effective surface stresses in areas where new crevasses open (e.g., cyan arrow in Fig. 6). This might also allow pinning down the threshold for the initiation of crevassing more precisely.**

*We appreciate this suggestion, and we have revised both the methods of selecting these metrics and the figures that include the crevasses metric. Using a 10% (excluding 10% of crevasses at lowest stress) and 90% (including 90% of crevasses) threshold, crevasses now generally occur between 82 and 165 kPa. Throughout the text, we now cite these values. One of the other reviewers also wanted a more thorough explanation of the upper bound on critical stress. They speculated that this is likely due to class imbalance at high stresses, and we agree. We've included histograms of stress across the basin to show the distributions of von Mises stresses for crevassed and uncrevassed locations. We've also changed the crevasse threshold in figure 7c. to be 82 kPa (consistent with the 10[th] percentile threshold we use to define the bounds).*

**Minor remarks:**

**L30: I do not think a new paragraph is needed here.**

*We have now not included the paragraph break.*

**L170-173: The trace spacing remains unclear without knowing the stacking. With no stacking, these numbers would give a very fine trace spacing of 1.6 mm. The horizontal resolution is also not directly set by the trace spacing, which only gives a lower bound, but it depends on the distance to the target and radar system characteristics.**

*Although additional processing is possible, we refer to values for the data as posted on the CReSIS public web server. Radar trace presuming (stacking) and SAR focusing result in data that are posted to a distance interval of 2.7 m. We've changed this section to clearly state these assumptions.*

**L227: n is not introduced.**

*We have introduced n more carefully in the preceding section. Thank you for the suggestion.*

**L294: The meaning of the upper 210 kPa envelope is unclear to me; from Fig. 8b, it seems that in general, no higher stresses are present at Thwaites Glacier. Do the 210 kPa only reflect this fact, or is there a deeper reason why higher stresses cannot build up, for example, because stresses are released by the formation of crevasses?**

*We have revised these calculations to be more quantitative. Following a previous reviewer suggestion, we now bound crevasses using two ellipses. 90% of crevasses occur above the lower bound ellipse, which corresponds to 82 kPa. 90% of the crevassed pixels occur below the upper bound ellipse, which corresponds to 165 kPa. These values were calculated including all velocity scenes and crevasse classifications (all quarterly images from 2015-2022, inclusive). To display the full range of values, we also plot histograms of the full range of von Mises stresses (see Figure 9). These values show that there are very few pixels with surface stresses that correspond to von Mises stresses greater than 210 kPa and no pixels greater than 250 kPa. From the data presented, we do not find stresses that exceed these values, but most of these high stress pixels are also associated with crevassing. The histograms of von Mises stress suggest that crevassing peaks for much lower stresses, so we suspect that 210-250 kPa is the highest stresses that occur, and not that crevassing prevents stresses from exceeding some critical values. The histograms do not indicate a single critical stress associated with crevassing, but rather a range of values centering near 125 kPa. The*

*velocities used to calculate the surface stresses assume a rheology representing strain that is uniform with depth.*

**L330-331: More specifically than saying off-nadir, these could be crevasses that initiate next to the radar line and are recorded from off-track directions. Interestingly, these reflections do not show a refraction shadow, which supports this interpretation.**

*We have added the observation that these features do not include a refraction shadow. We have also added an additional sentence to speculate on the nature of these features. In Sentinel imagery the crevasses appear to cross perpendicular to radar profiles, which would be inconsistent with off-nadir reflections.*

**L331-333: Is there evidence that such large flaws can form deep in the interior of a glacier? How could this be explained? Is it also an option that these are former crevasses that advected down from the upstream crevasse fields?**

*Large flaws like this have been observed elsewhere in the ice sheet. For instance, the impulse radar surveys along the Siple Coast ice streams in the 1990s and early 2000s. These have previously been linked to flaws/buried surface crevasses. The reviewer is correct in their interpretation that these formed upstream and likely advected downstream. This is clearly illustrated in Fig. 6, and we have added a sentence to the caption for that figure to explain this. Crevasses can form upstream particularly where ridges in basal shear stress promote locally high driving stresses that are coherent with variations in von Mises stress (Fig. 9, S8).*

**L346-345: Figures S8 and 5e do not exist.**

*This was a typo that we have corrected in the text.*

**L358-361: Is there other evidence that the dipping of internal layers into the crevasse is an actual signal that can be attributed to a disruption of the flow? Or could these dipping layers also be caused by off-nadir reflections, for example, from hyperbolic reflections from the layering at the sides of the crevasse or from large crevasses that are not perpendicularly oriented to the radar line?**

*You are right, that there is always some ambiguity in interpreting the geometry of scattering in a complex, 3D environment. There are a few reasons we think what we see are layers within the ice rather than off-nadir scattering that arrives at the same range and location as expected layering. Where we get scattering from the crevasse walls, we do see features that look like hyperbolic point diffractors, but they dip away*

*from the scattering source (and therefore, away from the crevasse void space). The layer dips that we interpret here dip toward the crevasse. They are also consistently traceable into the layering on either side (which suggests a nice, specular planar reflector, i.e., an internal layer). We think the most parsimonious explanation is that they are in fact the englacial layers.*

**L449-451: The discussion of the effect of crevasses on the firn air content is interesting, particularly the remark that it weakens the impact of horizontal divergence. However, I am not convinced by the estimated 2-8 m of unaccounted firn air content at lower Thwaites glacier due to crevasses. The total extent of crevasses cannot be directly identified by the crevassed area. For example, in Fig. 10c nearly the whole area between 7 km and 10.5 km distance has a high crevasse probability, but this of course only indicates the presence of many small crevasses in that region and not of a single big one.**

*We recognize that the apparent density of these features is difficult to measure with observations that integrate very near surface depth information (i.e. penetration depths of 10-15m as the reviewer noted earlier). There are also assumptions for the vertical depth of these features which make their volume difficult to constrain. We do think firn densification is likely enhanced by horizontal divergence (there are observations of this in NE Greenland where crevassing is not present; Christianson et al., 2014; Riverman et al., 2020); however, we speculate that in areas like Thwaites void space introduced by crevasses does need to be considered. The value of 2-8m was derived using a conservative estimate for the crevasse spacing density, not the estimates that one might infer from the imagery near the grounding zone of Thwaites. The depth that was assumed is similar (25-50 m) to the horizontal resolution of the imagery. More work clearly needs to be done on this topic. Our goal is to motivate this future work, with realistic estimates based on conservative assumptions that capture a signal we see in observations and have attempted to explain with idealized models.*

**Figure 4: The clarity of these figures could be improved by showing the radar lines (a-c) and the radargrams (d-e) in the same orientation, as it is done in Fig. 10. This also applies to Figs. S5-S7.**

*We have edited these figures and made the orientations the same (flow approximately to the left in all of these profiles). We've also changed the orientation of the plots to be consistent with the profiles.*

**Figure 8: Typo in "envelopes".**

*Thank you for noticing this. We've tried to go through and change instances of envelopes to ellipses*

**Figure 10: The orientation of the radar profiles is not indicated.**

*We have used a dot along the profile to indicate the start of the profile in mapview and the panel of the radargram. We've also indicated the flow direction of the radar profiles. All radar profiles are plotted in the same direction as presented in the maps. Ice always flows from right (inland) to left (seaward).*

**References**

Gerli, C., Rosier, S., & Gudmundsson, G. H. (2023). Activation of existing surface crevasses has limited impact on grounding line flux of Antarctic ice streams. Geophysical Research Letters, 50(6), e2022GL101687

Izeboud, M. and Lhermitte, S.: Damage detection on antarctic ice shelves using the normalised radon transform, Remote Sensing of Environment, 284, 113 359, https://doi.org/https://doi.org/10.1016/j.rse.2022.113359, 2023.

Jelitto, H. and Schneider, G.A., 2018. A geometric model for the fracture toughness of porous materials. Acta materialia, 151, pp.443-453. https://doi.org/10.1016/j.actamat.2018.03.018

Lai, C.-Y., Kingslake, J., Wearing, M. G., Chen, P.-H. C., Gentine, P., Li, H., Spergel, J. J., and van Wessem, J. M.: Vulnerability of Antarctica's ice shelves to meltwater-driven fracture, Nature, 584, 574–578, https://doi.org/10.1038/s41586-020-2627-8, 2020.

Surawy-Stepney, T., Hogg, A. E., Cornford, S. L., and Hogg, D. C.: Mapping Antarctic Crevasses and their Evolution with Deep Learning Applied to Satellite Radar Imagery, The Cryosphere Discussions, 2023, 1–32, https://doi.org/10.5194/tc-2023-42, 2023b.

---

## Referee Report (RR1)

**Review of "Inland migration of near-surface crevasses in the Amundsen Sea Sector, West Antarctica"**

Having read the revised manuscript along with the authors' diligent and comprehensive responses to reviewer comments, I am of the opinion that the article has been improved and should be published. I just have a couple of suggested revisions.

1. Line 64: "we extend deep-learning neural network frameworks to satellite synthetic aperture radar (SAR) imagery collected by the Sentinel-1 constellation from 2015–2022". Some of the studies you cite in this line use Sentinel-1 imagery, so I'm not sure the statement is quite accurate - maybe it's best to say you provide an alternative.

2. In my original review, I suggested removing the section on the firn model as it didn't add much to the paper, and was a slightly confusing presentation of a simple idea. The authors have responded that they'd like to keep it, which is fair enough. However, I think there are a few statements left in the that article that suggest something beyond a straightforward application of an existing model. I recommend tidying up these last bits, e.g. line 72: "We then develop a simple geometric model for the tensile strength of polar firn to aid the interpretation of our observations.". (I still think that the article would benefit from boiling a lot of the content around the firn model to a statement of the form "we apply the porous material model of Jelitto and Schnewider (2018) to firn.")

3. In general, I think the figures need a bit of tidying up. In particular, making sure text is large enough to read.

**References:**

Jelitto, H. and Schneider, G.A., 2018. A geometric model for the fracture toughness of porous materials. Acta materialia, 151, pp.443-453. https://doi.org/10.1016/j.actamat.2018.03.018

---

## Author Response (AR2)

**Response to Reviewer**

**Line 64: "we extend deep-learning neural network frameworks to satellite synthetic aperture radar (SAR) imagery collected by the Sentinel-1 constellation from 2015–2022". Some of the studies you cite in this line use Sentinel-1 imagery, so I'm not sure the statement is quite accurate - maybe it's best to say you provide an alternative.**

None of the papers that we cite in line 64 use SAR data, and in the next line we have already written: "Our results provide an alternative model framework to other published deep learning approaches (surawy-stepney et al., 2023a and surawy-stepney et. al., 2023b)."

**2. In my original review, I suggested removing the section on the firn model as it didn't add much to the paper, and was a slightly confusing presentation of a simple idea. The authors have responded that they'd like to keep it, which is fair enough. However, I think there are a few statements left in the that article that suggest something beyond a straightforward application of an existing model. I recommend tidying up these last bits, e.g. line 72: "We then develop a simple geometric model for the tensile strength of polar firn to aid the interpretation of our observations.". (I still think that the article would benefit from boiling a lot of the content around the firn model to a statement of the form "we apply the porous material model of Jelitto and Schnewider (2018) to firn.")**

We have adapted the model by Jeliitto and Schneider (2018), which is part of why we've described it in detail in the text. We assume different relationships between the volume of pore space in the firn and the firn structure. We can make this more clear though. We also think it strengthens the presentation of this work when we recognize that tensile weakening of porous materials is not a new idea and is well developed in literature on ceramics.

**3. In general, I think the figures need a bit of tidying up. In particular, making sure text is large enough to read**

We will work on this with the editorial team as part of sizing of the figures for the final layout of the manuscript. We appreciate this comment though.

**References**

Jelitto, H. and Schneider, G.: A geometric model for the fracture toughness of porous materials, Acta Materialia, 151, 443–453, https://doi.org/10.1016/j.actamat.2018.03.018, 2018.

Surawy-Stepney, T., Hogg, A. E., Cornford, S. L., and Davison, B. J.: Episodic dynamic change linked to damage on the Thwaites Glacier Ice Tongue, Nature Geoscience, 16, 37–43, https://doi.org/10.1038/s41561-022-01097-9, 2023a.

Surawy-Stepney, T., Hogg, A. E., Cornford, S. L., and Hogg, D. C.: Mapping Antarctic Crevasses and their Evolution with Deep Learning Applied to Satellite Radar Imagery, The Cryosphere Discussions, 2023, 1–32, https://doi.org/10.5194/tc-2023-42, 2023b.